# Learning to Move Before Learning to Do: Task-Agnostic pretraining for VLAs

**Junhao Shi** [1 2]  **Siyin Wang** [1 2]  **Xiaopeng Yu** [1 2]  **Li Ji** [1 2]  **Jingjing Gong** [2 †]  **Xipeng Qiu** [1 2 †]

## Abstract

Vision-Language-Action (VLA) models are bottlenecked by the scarcity of expert demonstrations—expensive triplets of observations, language instructions, and actions. We propose that learning "how to move" can be decoupled from learning "what to do," and that the former requires no task labels at all. Our two-stage framework, **Task-Agnostic Pretraining (TAP)** first pretrains on abundant, cheap *task-agnostic* data (discarded off-task trajectories or autonomous robot play) using an Inverse Dynamics objective that predicts actions from consecutive observations. This self-supervised phase instills physical affordances—grasping, contact dynamics, end-effector control—without human annotation. A lightweight second stage then aligns these physical priors with language instructions using minimal expert data. On the SIMPLER benchmark, our approach matches models trained on 1M+ expert trajectories while using orders of magnitude less labeled data, achieving a 10% absolute gain over standard behavior cloning. In real-world WidowX experiments, it surpasses internet-scale baselines under visual distribution shifts (e.g., 25% vs. 0% under camera perturbations), demonstrating that task-agnostic pretraining yields robust, transferable physical representations for Embodied AI.

## 1. Introduction

The development of Vision-Language-Action (VLA) models (Intelligence et al., 2025; Kim et al., 2024; Team et al., 2024; Collaboration et al., 2023; Black et al., 2024) has opened promising avenues for building general-purpose robots. However, training these models requires an enormous amount of high-quality robotic data. The predominant approach to acquiring such data relies on human teleoperation, where expert operators guide robot movements while providing language based task annotations (Padalkar et al., 2023; Walke et al., 2023; Khazatsky et al., 2024a). This paradigm is inherently *passive* from the robot's perspective: the robot serves merely as a vessel for capturing human demonstrations, contributing no exploration of its own. Beyond being prohibitively expensive and labor-intensive, this data collection scheme is fundamentally *unnatural*—it conflates the embodied experience of a robot with the disembodied intentions of a human operator. Crucially, it is also *non-scalable*: the rate of data acquisition is bottlenecked by the availability and endurance of human operators, making it impractical to collect the diverse, large-scale datasets required for truly general-purpose manipulation.

Consider, by contrast, how biological agents acquire motor competence. A human infant does not learn to grasp, manipulate, and interact with objects by passively receiving demonstrations from an expert. Instead, infants engage in spontaneous, *task-agnostic* exploration—reaching, touching, dropping, and observing the consequences of their actions (Hoch et al., 2019). This curiosity-driven self-exploration is philosophically *task-unaware*: the infant is not optimizing for any specific goal but is rather building an internal model of how the world responds to its actions (Kidd & Hayden, 2015). Through this process, the infant develops a rich understanding of physics, affordances, and sensorimotor contingencies (Adolph & Hoch, 2019) long before any explicit task instruction is given. The grounding in "how to move" emerges naturally from active interaction, decoupled from "what to do."

Inspired by this developmental perspective, we argue that robots should similarly benefit from *active*, task-agnostic data collection. Such data is abundant and inexpensive to acquire: robots can autonomously generate vast amounts of interaction trajectories through random play, without human supervision or task-specific annotations. Yet, this valuable resource remains largely underutilized in current VLA training pipelines, which discard any trajectory that lacks explicit task instructions. We observe that while task instructions are necessary for learning "what to do," they are not required for learning "how to move"—the fundamental dynamics and physical affordances of manipulation.

---

[†]Corresponding Authors. [1]Fudan University, Shanghai, China. [2]Shanghai Innovation Institute, Shanghai, China. Correspondence to: Junhao Shi <24110240071@m.fudan.edu.cn>.

*Proceedings of the 43$^{rd}$ International Conference on Machine Learning*, Seoul, South Korea. PMLR 306, 2026. Copyright 2026 by the author(s).

In this work, we demonstrate how to unlock the value of task-agnostic data for VLA learning. We identify two abundant sources of such data: (1) *task-irrelevant trajectories*—existing demonstrations collected for unrelated tasks that are typically discarded, and (2) *autonomous random play*—interaction data generated by robots exploring their environment without human supervision. To extract physical knowledge from this unlabeled data, we employ an Inverse Dynamics objective (Brandfonbrener et al., 2023), where the model learns to predict the action $a_t$ required to transition from observation $o_t$ to a future state $o_{t+1}$. This self-supervised formulation forces the model to attend to dynamic elements—end-effector motion, object displacement—while ignoring static background noise, thereby acquiring "physical common sense" without any language annotations. With this grounding in place, only a minimal set of expert demonstrations is needed to align the learned affordances with task-specific linguistic instructions. We call this approach **Task-Agnostic Pretraining (TAP)**.

We evaluate our approach on both the Simpler benchmark (Li et al., 2024) and real-world WidowX 250s robot experiments. In Simpler, by repurposing task-irrelevant trajectories from the Bridge dataset (Walke et al., 2023)for inverse dynamics training, we significantly improve performance on downstream tasks compared to standard training baselines. In the real world, we demonstrate that pretraining on autonomously generated random trajectories reduces the dependency on expensive expert teleoperation. Our results show that our method achieves superior sample efficiency and generalization, effectively transforming "useless" task-agnostic data—collected in the spirit of infant-like self-exploration—into a valuable resource for scaling Embodied AI.

## 2. Related Works

### 2.1. Large-scale pretraining for Vision-Language-Action Models

For years, visuomotor control was dominated by task-specific policies trained within constrained environments. While effective for isolated skills, these methods struggled to generalize to novel objects or unstructured language instructions(Ma et al., 2026; Zhang et al., 2025; Zhong et al., 2025).

Inspired by the scaling laws of natural language processing, RT-1 (Brohan et al., 2023) and RT-2(Zitkovich et al., 2023)pioneered the shift toward "generalist" agents, demonstrating that unifying perception, language, and action into a single Transformer and scaling data diversity could unlock emergent generalization capabilities. To harness the scaling potential of these evolving architectures, the community has increasingly focused on aggregating massive,

multi-embodiment datasets. This effort culminated in the Open X-Embodiment (OXE) (Collaboration et al., 2023) dataset, which unifies diverse data sources such as Bridge-Data (Walke et al., 2023) and DROID (Khazatsky et al., 2024b). Fueled by these millions of expert-teleoperated trajectories, the latest generation of VLA models has achieved remarkable success. Systems such as OpenVLA (Kim et al., 2024), $\pi_0$ (Black et al., 2024), $\pi_{0.5}$ (Intelligence et al., 2025), and Gen-0 (Team, 2025) integrate internet-scale VLM backbones with diffusion or flow-matching action heads, achieving unprecedented levels of dexterity and real-time control through large-scale pretraining. Furthermore, RoboOmni (Wang et al., 2025) extends this paradigm by exploring and expanding the native end-to-end omni-modal capabilities of VLAs.

Despite their impressive performance, current state-of-the-art VLAs still suffer from a fundamental bottleneck: the "data wall." Their generalization is strictly bounded by the scale of expert human demonstrations, where generalization is constrained by the prohibitive cost of scaling expert demonstrations. Our work challenges this brute-force scaling regime by pretraining manipulation priors using cheap, task-agnostic data, thereby significantly reducing the dependency on expensive expert demonstrations.

### 2.2. Dynamics Learning in Robotics

Dynamics learning equips robot policies with physical reasoning by modeling state transitions, typically categorized into *forward dynamics* ($\hat{s}_{t+1} \leftarrow f_{\text{fwd}}(s_t, a_t)$) and *inverse dynamics* ($\hat{a}_t \leftarrow f_{\text{inv}}(s_t, s_{t+1})$). Prior works have extensively leveraged these objectives to pretrain visual representations. Explicit modeling approaches, such as MIDAS (Rendall et al., 2025), SMART (Sun et al., 2023), and PACT (Bonatti et al., 2023), directly predict future states or actions to capture local physical laws and environmental transitions. Conversely, implicit methods like Vi-PRoM (Jing et al., 2023) and MaskDP (Liu et al., 2022) internalize dynamics through temporal reordering or masked reconstruction tasks without explicit state prediction. Furthermore, video-based frameworks such as VPT (Baker et al., 2022) and GR-1 (Wu et al., 2024) scale these objectives to large unlabeled datasets, utilizing inverse dynamics primarily for pseudo-labeling internet videos or anticipating future frames to refine action prediction.

Most prior methods treat dynamics learning either as an auxiliary objective or a tool for pseudo-labeling data. In contrast, we employ inverse dynamics as a *standalone pretraining phase* specifically to unlock the value of massive, task-agnostic action data. By learning physical priors—such as object affordances and kinematics—*before* encountering any task semantics, our method provides a robust structural foundation that significantly enhances downstream learning

efficiency and performance.

# 3. Task-Agnostic Data for Physical Grounding

The fundamental bottleneck in VLA learning is the scarcity of aligned triplets $(o, l, a)$—observations paired with both language instructions and expert actions. We propose to bypass this bottleneck by exploiting a vastly more abundant resource: **task-agnostic interaction data**. Our key insight is a *Decomposition Hypothesis*: action generation can be factorized into (1) perceiving physical affordances ("how to move") and (2) grounding semantic intent ("what to do"). Crucially, the former can be learned entirely from task-agnostic data, without any language annotations.

As shown in Figure 1, our framework operationalizes this hypothesis by first harvesting massive task-agnostic data (Sec. 3.1) to learn physical priors via Inverse Dynamics (Sec. 3.2), and subsequently aligning these priors with semantic instructions via finetuning (Sec. 3.3).

## 3.1. Harnessing Task-Agnostic Data

We define **task-agnostic data** as any robot interaction trajectory $\tau = (o_0, a_0, o_1, a_1, \ldots, o_T)$ that lacks explicit task semantics—i.e., no language instruction $l$ is associated with the trajectory. Such data captures valid physical interactions (the robot moved, objects responded) but carries no human-defined "purpose."

**Sources of Task-Agnostic Data.** This data is abundant, cheap, and exists in two primary forms:

- **Repurposed Existing Datasets.** Large-scale robotic datasets (e.g., BridgeData (Walke et al., 2023), Open X-Embodiment (Collaboration et al., 2023)) contain thousands of trajectories collected for tasks *irrelevant* to a target deployment. For instance, if the target downstream task is "put carrot on plate," trajectories of "open drawer" or "wipe table" are traditionally discarded. We argue these contain rich physical priors—grasping dynamics, collision responses, end-effector control—that transfer across tasks.

- **Autonomous Random Play.** Robots can generate unlimited interaction data through self-supervised exploration. By executing randomized end-effector commands, the robot pushes, sweeps, topples, and grasps objects without any human involvement. This "play" data is virtually free to collect and captures the robot's specific embodiment and workspace.

**Autonomous Collection Pipeline.** To ensure the robot collects meaningful, contact-rich data safely, we procedurally generate trajectories grounded in a verified spatial prior. First, an operator teleoperates the robot without specific

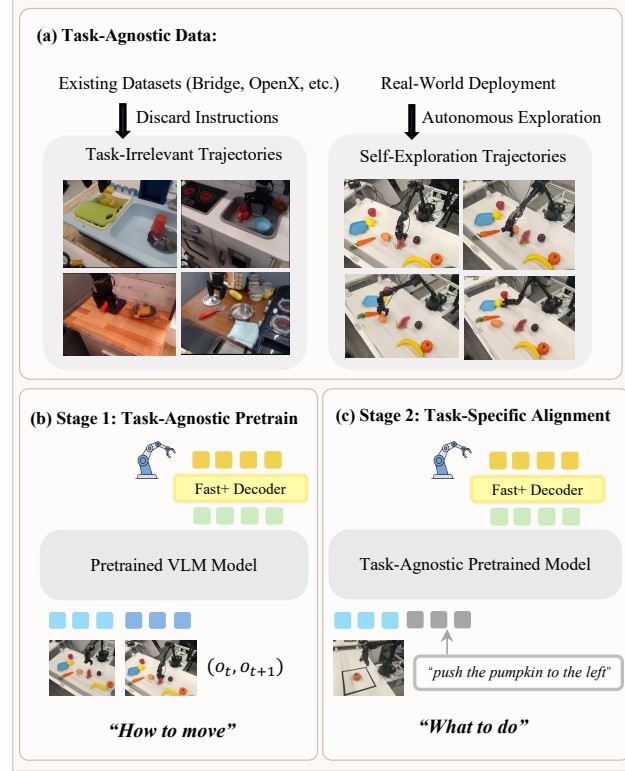

*Figure 1.* **Overview of the Proposed Task-Agnostic Pretraining (TAP) Framework.** **(a) Data Sources:** We leverage massive amounts of cheap, unlabeled interaction data—sourced from existing heterogeneous datasets (e.g., Bridge) or autonomous robot self-exploration — discarding any original task labels. **(b) Stage 1 (Task-Agnostic Pretraining):** The model is pretrained using a self-supervised Inverse Dynamics (ID) objective. By predicting the action $a_t$ required to transition between frames $o_t$ and $o_{t+1}$, the model learns robust physical affordances and motor control ("how to move") without human supervision. **(c) Stage 2 (Task-Specific Alignment):** The pretrained model is then finetuned on a small set of language-annotated expert demonstrations. This stage aligns the prelearned physical priors with high-level semantic instructions ("what to do"), achieving high performance with significantly improved data efficiency.

tasks to densely cover the reachable workspace. We filter this data and apply Voxel Grid Downsampling to construct a uniform, discrete safe pose library $\mathcal{P}_{safe}$. Next, we stochastically sample waypoints from this library to form continuous trajectories. To prevent the end-effector from hovering and force meaningful interactions (e.g., pushing, sliding), we apply a contact heuristic that forces a descent if the trajectory remains above an elevation threshold ($z_{thresh}$) for too long. Finally, boundary-aware Gaussian noise is injected to maximize diversity before the robot executes the trajectory.ng). Detailed pipeline setups are provided in Appendix A.

**Algorithm 1** Constrained Procedural Trajectory Generation

---

**Require:** Raw teleoperation poses $\mathcal{P}_{raw}$, safety bounds $\mathcal{B}$, voxel size $v_{size}$, min distance $d_{min}$, elevation threshold $z_{thresh}$, max high-elevation steps $c_{max}$, noise scale $\sigma$.
**Ensure:** Procedural trajectory dataset $\mathcal{D}_{play}$

1: **Phase 1: Safe Pose Library Initialization**
2: $\mathcal{P}_{valid} \leftarrow \{p \in \mathcal{P}_{raw} \mid p \in \mathcal{B}\}$
3: $\mathcal{P}_{safe} \leftarrow \text{VoxelGridDownsample}(\mathcal{P}_{valid}, v_{size})$
4:
5: **Phase 2: Autonomous Data Collection**
6: $\mathcal{D}_{play} \leftarrow \emptyset$
7: **while True do**
8: $\quad \mathcal{W} \leftarrow \text{SampleWaypoints}(\mathcal{P}_{safe}, d_{min})$ {{Stochastic sampling}}
9: $\quad \mathcal{W}_{contact} \leftarrow \text{ContactHeuristic}(\mathcal{W}, z_{thresh}, c_{max})$ {{Bound consecutive high-Z points}}
10: $\quad \tau \leftarrow \text{CosineInterpolate}(\mathcal{W}_{contact})$
11: $\quad \tau \leftarrow \text{Clip}(\tau + \mathcal{N}(0, \sigma), \mathcal{B})$ {{Inject boundary-aware exploration noise}}
12: $\quad$ Execute $\tau$ and append recorded transitions to $\mathcal{D}_{play}$
13: $\quad$ **if** human intervention triggered (e.g., $\Delta t \geq 30$ mins) **then**
14: $\quad\quad$ Shuffle or swap objects in the workspace
15: $\quad$ **end if**
16: **end while**
17: **return** $\mathcal{D}_{play}$

---

## 3.2. Stage 1: Task-Agnostic Pretraining

To extract physical knowledge from unlabeled trajectories, we formulate a self-supervised **Inverse Dynamics (ID)** objective. Given two observations $(o_t, o_{t+1})$, the model predicts the action $a_t$ that caused the transition:

$$p(a_t \mid o_t, o_{t+1}) \tag{1}$$

**Why Inverse Dynamics?** Predicting the action $a_t$ that caused a state transition requires the model to focus on *what changed* between frames—the motion of the end-effector and the displacement of manipulated objects—while ignoring static background elements (lighting, textures, clutter). This forces the visual encoder to learn *dynamics-aware* representations that encode "how the world changes" rather than "how the world looks."

**Training Objective.** We instantiate $f_\theta$ using a Vision-Language Model (VLM) backbone. During inverse dynamics training, we construct a visual-only input sequence by treating the future observation $o_{t+1}$ as an implicit *visual goal*. Let $\phi : \mathcal{O} \to \mathbb{R}^{L \times d}$ denote a visual encoder that maps an observation to a sequence of $L$ tokens of dimension $d$.

We optimize the model parameters $\theta$ by minimizing the mean squared error between predicted and ground-truth actions:

$$\hat{a}_t \leftarrow f_\theta(\phi(o_t), \phi(o_{t+1})) \tag{2}$$

$$\mathcal{L}_{\text{ID}}(\theta) = \mathbb{E}_{(o_t, a_t, o_{t+1}) \sim \mathcal{D}_{\text{TAP}}} \left[ \|\hat{a}_t - a_t\|_2^2 \right] \tag{3}$$

Upon convergence, the model has acquired robust physical priors—spatial reasoning, affordance detection, and motor coordination—purely from task-agnostic interactions, without any language supervision.

## 3.3. Stage 2: Task-Specific Alignment

Once the model understands "how to move," we align this knowledge with "what to do" using a minimal set of expert demonstrations $\mathcal{D}_{\text{expert}} = \{(o_t, l, a_t)\}_{t=1}^{N_{\text{expert}}}$, where each sample includes a language instruction $l \in \mathcal{L}$.

**Input Representation.** The input structure shifts from visual-goal conditioning to language-instruction conditioning. Let $\psi : \mathcal{L} \to \mathbb{R}^{M \times d}$ denote a text encoder that maps a language instruction to a sequence of $M$ tokens. The model now receives:

$$\hat{a}_t \leftarrow f_\theta(\phi(o_t), \psi(l)) \tag{4}$$

**Why Does This Work?** Although the conditioning signal changes from a future observation $o_{t+1}$ to a language instruction $l$, the backbone $f_\theta$ and action head are reused. The model has already learned to map visual contexts to motor outputs during the task-agnostic phase. This stage essentially learns a lightweight projection from semantic space to the pre-established dynamics space—requiring significantly fewer labeled samples than training from scratch.

**Training Objective.** The model is finetuned via standard behavior cloning:

$$\mathcal{L}_{\text{BC}}(\theta) = \mathbb{E}_{(o_t, l, a_t) \sim \mathcal{D}_{\text{expert}}} \left[ \|f_\theta(\phi(o_t), \psi(l)) - a_t\|_2^2 \right] \tag{5}$$

# 4. Experiments

We design our experiments to verify whether self-supervised physical priors can effectively bypass the expert data bottleneck. We structure our analysis around three hypothesis-driven questions:

**RQ1 (Effectiveness & Efficiency):** Can task-agnostic interaction data, combined with inverse dynamics pretraining, match or exceed the performance of models trained on massive expert datasets while using significantly less labeled data?

**RQ2 (Mechanism):** Does task-agnostic pretraining improve low-level physical affordances (e.g., grasping, contact), as evidenced by sub-goal success rates and learned visual representations?

*Table 1.* **Comparison of Training Paradigms and Data Scale.** Traditional foundational models rely on massive, expensive, task-labeled expert demonstrations. In contrast, our approach leverages cheap, task-agnostic data via a self-supervised Inverse Dynamics (ID) objective.

| Model | Pretraining Data | Data Scale | Objective (Labels Required) |
|---|---|---|---|
| RT-1-X (Collaboration et al., 2023) | Open X-Emb. | ~1.0M | BC (Yes) |
| OpenVLA (Kim et al., 2024) | Open X-Emb. | ~970k | BC (Yes) |
| Nora (Hung et al., 2025) | Open X-Emb. | ~1.0M | BC (Yes) |
| Octo (Team et al., 2024) | Open X-Emb. | ~800k | Masked BC (Yes) |
| $\pi_0$ (Black et al., 2024) | Multi-Emb. | Massive | BC (Yes) |
| **TAP (Stage 1)** | Task-Agnostic (Irrelevant Bridge / Self-Exploration) | **20k** (Sim) / **30h** (Real) | **Inverse Dyn. (No)** |
| **TAP (Stage 2)** | Task-Specific Expert Data | **5k** (Sim) / **0.2k** (Real) | BC (Yes) |

**RQ3 (Robustness):** Does pretraining on diverse, autonomous exploration data improve resilience to real-world distribution shifts, including visual perturbations and environmental clutter?

### 4.1. Experimental Setup

To rigorously evaluate our Decomposition Hypothesis, we benchmark TAP across both simulated (SIMPLER (Li et al., 2024)) and real-world (WidowX 250) environments.

**Model & Baselines.** We instantiate our framework using a Qwen2.5-VL (3B) backbone coupled with a SigLIP visual encoder. As detailed in Table 1, we compare TAP against two distinct categories of baselines. It is crucial to clarify the intended role of each:

(1) **Standard BC**: An identical architecture trained from scratch purely on limited expert data. *This serves as our primary experimental comparison* to rigorously isolate and prove the value of task-agnostic pretraining.

(2) **Large-scale Pretrained VLAs**: SOTA foundational models (OpenVLA, NORA, Octo, and $\pi_0$) pretrained on the massive Open X-Embodiment (OXE)(Collaboration et al., 2023) dataset. OXE comprises over 1 million expert-teleoperated, language-annotated trajectories—representing an enormous investment of human labor and annotation cost. By contrast, TAP relies merely on 30 hours of autonomous random play requiring minimal human effort. Therefore, we include these VLAs not as direct competitors, but as *approximate upper-bound references* to demonstrate what internet-scale, high-quality expert pretraining can achieve.

**Action Representation.** We adopt a **delta-pose end-effector action space**, where $a_t \in \mathbb{R}^7$ encodes the relative position change $(\Delta x, \Delta y, \Delta z)$, orientation change (represented as a 3D axis-angle vector), and a scalar gripper command. Predicting relative motion rather than absolute poses enables the model to learn local interaction dynamics that are invariant to global workspace coordinates—a property critical for transferring physical priors across different robot configurations.

**Evaluation Protocols.** In simulation, we evaluate four distinct manipulation tasks, reporting success rates averaged over 50 episodes per checkpoint. In the real world, we conduct over 600 physical trials across five testing conditions (spanning from in-domain setups to severe out-of-distribution scenarios) to probe the boundaries of model robustness. Detailed implementation specifics are provided in Appendix B.4.

### 4.2. Simulation Results: Effectiveness and Physical Grounding

To address **RQ1** and **RQ2**, Table 2 breaks down performance into *Partial* success (successful grasping) and *Entire* success (full task completion, including precise placement) across four manipulation tasks.

**RQ1: Effectiveness and Efficiency.** Our TAP-20k model achieves an Avg-All success rate of 33.32%, significantly outperforming massive foundational models like OpenVLA (7.75%) and RT-1-X (3.03%). Notably, these large-scale models frequently suffer from a 0% Entire success rate on complex tasks like *Spoon on cloth* or *Eggplant in Basket*, indicating severe cross-embodiment degradation when fine-tuned on limited data. When compared against the Standard BC baseline (23.15%)—which shares an identical architecture and Stage 2 dataset—our pretraining yields a +10% absolute gain in Avg-All performance. Furthermore, Table 2 reveals a monotonic improvement across TAP's pretraining trajectory (from 8k to 20k episodes), rigorously confirming that deeper task-agnostic physical exposure directly translates to higher downstream task proficiency.

**RQ2: Mechanism via Partial Success Analysis.** Analyzing the Partial versus Entire metrics illuminates *how* TAP drives these improvements and overcomes the fundamental behavioral bottleneck. Our model registers an Avg-Partial success of 45.82%, mirroring Octo (42.30%) and approaching $\pi_0$ (53.10%). Partial success heavily depends on low-level physical competencies—end-effector alignment, precision reaching, and stable grasping—which are entirely independent of high-level task semantics.

This decomposition strongly validates our core hypothesis: Stage 1 pretraining successfully teaches the model **"how to move"** by coercing the visual encoder to infer actions from unlabeled observation pairs $(o_t \rightarrow o_{t+1})$. The model inter-

*Table 2.* **Success Rates on the SIMPLER Benchmark (WidowX Environment).** We report task-specific success rates for intermediate sub-goals (*Part.*) and full task completion (*Ent.*). Summary metrics include **Avg-Partial** (mean success rate of object grasping), **Avg-Entire** (mean full completion), and **Avg-All** (aggregate mean). The top two performances are in bold. **Fine-tuning Fairness & Context:** To ensure a rigorous evaluation of task-specific adaptation, OpenVLA, NORA, and $\pi_0$ were fine-tuned using the *exact same subset* of Stage 2 expert data as our TAP model (i.e., 5k trajectories for simulation). Conversely, the results for RT-1-X and Octo were directly cited from the original SIMPLER benchmark paper (Li et al., 2024) to provide a broader context. Overall, our task-agnostic pretraining significantly boosts physical grounding, frequently matching or exceeding foundational models trained on 1M+ trajectories.

| Type | Model Name | Spoon on cloth | | Carrot on plate | | Stack Blocks | | Eggplant in Basket | | Avg-Partial | Avg-Entire | Avg-All |
|---|---|---|---|---|---|---|---|---|---|---|---|---|
| | | Part. | Ent. | Part. | Ent. | Part. | Ent. | Part. | Ent. | | | |
| Reference | RT-1-X (Collaboration et al., 2023) | 4.2% | 0.0% | 16.7% | 0.0% | 0.0% | 0.0% | 3.3% | 0.0% | 6.05% | 0.00% | 3.03% |
| | OpenVLA (Kim et al., 2024) | 4.1% | 0.0% | 33.0% | 0.0% | 12.5% | 0.0% | 8.3% | 4.1% | 14.48% | 1.03% | 7.75% |
| | Nora (Hung et al., 2025) | 37.5% | 16.7% | 48.0% | 0.0% | 41.7% | 12.5% | 4.17% | 0.0% | 32.84% | 7.29% | 20.06% |
| | Octo (Team et al., 2024) | **50.0%** | 33.0% | **50.0%** | 25.0% | 29.2% | 0.0% | **40.0%** | 23.3% | 42.30% | 20.33% | 31.31% |
| | $\pi_0$ (Black et al., 2024) | 45.8% | 29.1% | 25.0% | 0.0% | 50.0% | 16.7% | **91.6%** | **62.5%** | **53.10%** | **27.05%** | **40.08%** |
| Baseline | Standard BC | 41.7% | 33.3% | 48.0% | 8.0% | 37.5% | 16.7% | 0.0% | 0.0% | 31.79% | 14.50% | 23.15% |
| Ours | **TAP-8k episodes** | **50.0%** | **37.5%** | 37.5% | 8.3% | **58.3%** | 4.2% | 0.0% | 0.0% | 36.45% | 12.50% | 24.47% |
| | **TAP-14k episodes** | 41.7% | 33.3% | **50.0%** | **16.7%** | **83.3%** | 12.5% | 4.2% | 0.0% | 44.80% | 15.62% | 30.21% |
| | **TAP-20k episodes** | **66.7%** | **58.3%** | **50.0%** | 0.0% | **58.3%** | **16.7%** | 8.3% | 8.3% | **45.82%** | **20.82%** | **33.32%** |

nalizes fine-grained motor control before ever encountering a language instruction. In standard pipelines, if the policy fails at the initial physical affordance bottleneck (e.g., dropping the object), semantic execution becomes impossible. TAP explicitly solves this bottleneck. By establishing a deep physical grounding during Stage 1, the model's representational capacity is entirely freed to focus on semantic goals (e.g., navigating to the specific target plate) during Stage 2 finetuning, seamlessly converting high Partial success into superior Entire success.

### 4.3. Real-World Experiments: Robustness via Self-Exploration

To answer **RQ3**, we evaluate our method on a physical WidowX 250s robot. As shown in Figure 2, we select two complementary tasks that probe distinct aspects of physical understanding:

- **Put the carrot on the plate** tests precision grasping in a classic pick-and-place scenario. Success requires the robot to grasp the carrot and release it stably onto the plate, evaluating *geometric alignment* and *grasping affordance*.

- **Push the pumpkin to the left** tests dynamic, non-prehensile manipulation. We define a $30 \times 30$ cm$^2$ arena with the pumpkin initialized at center ($\pm 1$cm). A trial succeeds only if the robot pushes the object such that more than 50% of it exits the boundary in the specified direction. Unlike grasping, pushing demands *sustained contact* and an implicit understanding of *object dynamics*—the spherical pumpkin will spin and deviate if force is not applied precisely through its center.

To rigorously assess out-of-distribution robustness, we establish a tiered evaluation protocol comprising four distinct

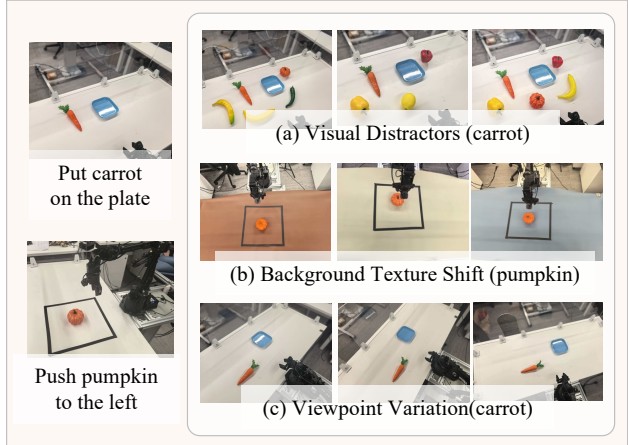

*Figure 2.* **Real-World Evaluation Setup and Robustness Protocols.** We evaluate our method on a physical WidowX 250 robot across two manipulation tasks: *Put carrot on plate* and *Push pumpkin*. To rigorously quantify resilience to distribution shifts (RQ3), we introduce systematic environmental perturbations: **(a) Visual Distractors** introduce unseen clutter (e.g., diverse fruits) to test semantic attention mechanisms; **(b) Background Texture Shifts** alter surface materials to evaluate visual invariance; and **(c) Viewpoint Variations** perturb camera extrinsics to assess geometric generalization. Combined with **Initial State Perturbations**, these scenarios probe the boundaries of model robustness beyond the training distribution.

categories of environmental variation: *Initial State Perturbations*, *Visual Distractors*, *Background Texture Shifts*, and *Viewpoint Variations*. These perturbations are designed to simulate real-world unpredictability and verify whether the policy relies on robust physical priors rather than spurious visual correlations. Detailed configurations for each scenario are provided in Appendix B.4.

**Data Collection & Baselines.** To simulate severe data scarcity, we strictly limit human supervision to only 200 expert trajectories per task. For Stage 1, the robot au-

*Table 3.* **Real-World Evaluation Success Rates (%).** Models are trained on only 200 expert demonstrations. Our TAP model is pretrained on 30 hours of autonomous self-exploration. **Bold** indicates the best performance. TAP demonstrates remarkable resilience, surpassing the internet-scale NORA baseline in dynamic tasks with clutter ("Visual Distractors") and consistently outperforming all baselines under severe visual perturbations ("Background Texture Shift" and "Viewpoint Variation").

| Evaluation Scenario | Task: Put the carrot on the plate | | | Task: Push the pumpkin to the left | | |
| --- | --- | --- | --- | --- | --- | --- |
| | From scratch | **TAP (Ours)** | NORA (SOTA) | From scratch | **TAP (Ours)** | NORA (SOTA) |
| **Standard Setup** | 20% | 40% | **65%** | 55% | 75% | **85%** |
| **Initial State Perturbation** | 20% | 30% | **65%** | 45% | 75% | **80%** |
| **Visual Distractors** | 5% | 30% | **40%** | 5% | **65%** | 60% |
| **Background Texture Shift** | 0% | **25%** | 10% | 0% | **65%** | 55% |
| **Viewpoint Variation** | 0% | **15%** | 0% | 0% | **25%** | 0% |
| **Average** | 9% | 28% | **36%** | 21% | **61%** | 56% |

tonomously collects 30 hours of task-agnostic random play data (Algorithm 1). We compare TAP against the **Standard BC** (trained from scratch) and **NORA** (Hung et al., 2025) (finetuned from massive OXE pretraining).

**RQ3: Robustness from Self-Exploration.** The results in Table 3 highlight two critical advantages of our framework, dissecting how physical priors combat real-world unpredictability:

**1) Overcoming Spurious Correlations in Clutter.** While NORA dominates in clean, standard setups (e.g., 85% on pushing), TAP exhibits vastly superior adaptability in chaotic environments. In the cluttered pushing task equipped with unseen fruits, standard BC drops to a near-random 5%, and NORA decays to 60%. In contrast, TAP maintains a 65% success rate. This indicates that without localized physical pretraining, policies easily overfit to spurious visual correlations in the background. TAP's self-exploration phase forces the model to attend to *causal interactive dynamics* (the relationship between the gripper and the manipulated object), rendering static visual distractors semantically invisible.

**2) Resilience to Severe Spatial and Textural Shifts.** The divergence is most profound under structural perturbations. When camera extrinsics are shifted significantly, NORA and the Standard BC suffer catastrophic spatial misalignment, frequently grasping at empty space (0% success on both tasks). Conversely, TAP retains robust functionality (15% and 25% success). Background texture shifts follow a similar pattern: replacing the wooden table with a colored cloth causes NORA's pushing performance to plummet to 55%, while TAP remains highly robust at 65%.

**Overall Performance & The Value of TAP.** Averaged across all demanding scenarios, TAP rivals or exceeds the overall success rate of the NORA baseline (e.g., 61% vs. 56% in the pushing task). Against this approximate upper-bound reference—which required a massive investment of over a million human-annotated trajectories—the fact that

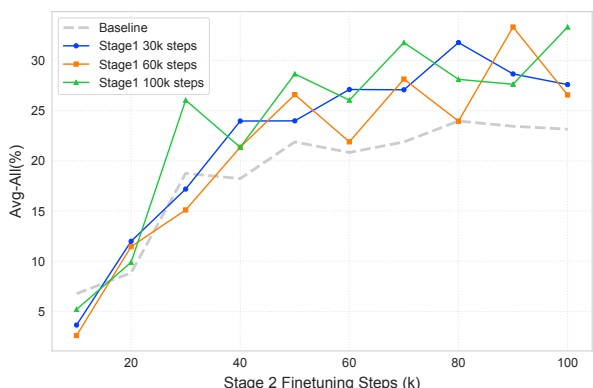

*Figure 3.* **Convergence Dynamics.** Avg-All success rates during Stage 2 finetuning. Initial learning rates are comparable across methods, but the Baseline (dashed) plateaus early while pretrained models (solid) achieve substantially higher final performance—demonstrating that task-agnostic pretraining expands learning capacity rather than accelerating convergence.

TAP achieves comparable overall proficiency and strictly superior robustness in out-of-distribution environments, utilizing merely 30 hours of autonomous random play, definitively proves the immense potential and scalability of task-agnostic physical pretraining.

## 4.4. Ablation Study: How Task-Agnostic Data Shapes Learning

Having established the effectiveness of our approach, we now investigate the *mechanisms* through which task-agnostic data improves downstream performance. We analyze three complementary aspects: convergence dynamics, data scaling behavior, and learned visual representations.

**Overcoming Early Saturation.** A fundamental limitation of standard Behavior Cloning is early saturation—models trained on limited expert data plateau quickly, unable to generalize beyond memorized trajectories. Figure 3 tracks validation success rates throughout Stage 2 finetuning.

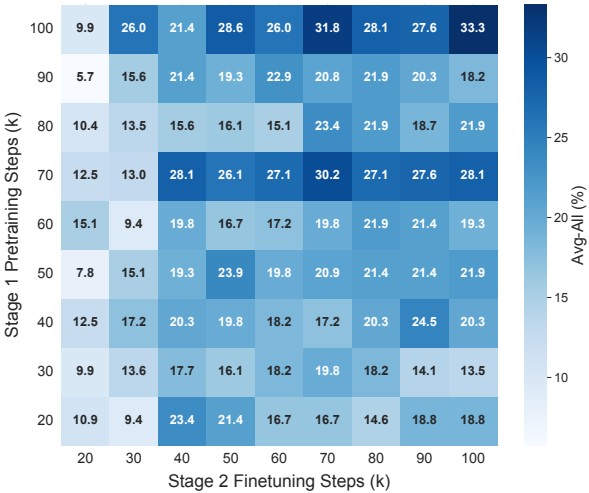

*Figure 4.* **Data Scaling Analysis for Overall Success (Avg-all).**
Avg-All success rates across varying Stage 1 (pretraining) and
Stage 2 (finetuning) durations. The upward gradient indicates that
pretraining scale sets the performance ceiling—extended finetuning cannot compensate for insufficient task-agnostic exposure.

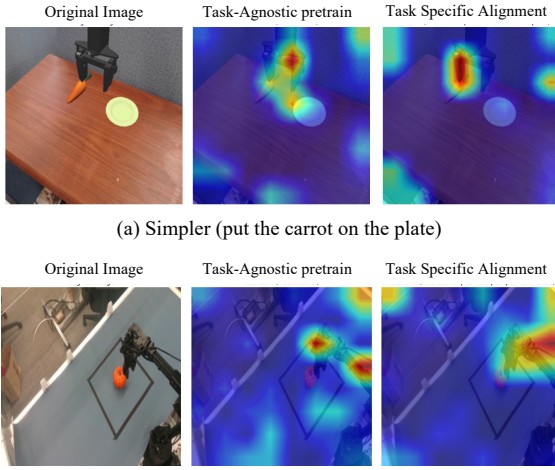

(a) Simpler (put the carrot on the plate)

(b) Real World (push the pumpkin to the left, Unseen background)

*Figure 5.* **Attention Map Analysis.** Comparison across simulation
(top) and real-world deployment (bottom). *Middle:* Task-agnostic
pretraining yields attention focused on manipulable entities (gripper, objects) without any language input. *Right:* Instruction tuning
refines attention toward robotic agents (grippers). The consistency
across domains confirms that learned physical priors transfer robustly to novel environments.

Notably, pretrained models (solid lines) and the Baseline
(dashed) exhibit similar initial learning rates, indicating that
task semantics are acquired at comparable speeds. The
critical divergence occurs later: while the Baseline saturates around 23% and oscillates, pretrained models continue
climbing to exceed 30%. This pattern suggests that task-agnostic pretraining does not speed up the acquisition of
task semantics—instead, it raises the upper bound on achievable performance. The physical priors acquired in Stage
1 prevent convergence to poor local optima, enabling the
model to extract more value from identical expert data.

**Data Scaling: A Necessary Foundation.** We next examine
how volumes of task-agnostic pretraining (Stage 1) and task-specific finetuning (Stage 2) jointly determine performance.
Figure 4 presents a systematic sweep across both axes.

The heatmap reveals a strict dependency: *downstream performance is bounded by pretraining scale*. With minimal
Stage 1 exposure (20k steps), extending Stage 2 yields diminishing returns—performance stagnates near 18% regardless of finetuning duration. This suggests that without sufficient diversity in task-agnostic interactions, the model lacks
the generalizable physical representations required for robust manipulation. Conversely, scaling Stage 1 to 100k steps
unlocks success rates exceeding 30%. The optimal regions
(dark blue) confirm that abundant task-agnostic data acts
as a regularizer, preventing overfitting to the limited expert
trajectories available during finetuning.

**Visualizing Learned Physical Priors.** To verify that task-agnostic data instills meaningful physical understanding,
we visualize Grad-CAM (Selvaraju et al., 2017) attention

maps from the model's final layer (Figure 5). We compare
attention patterns after Stage 1 (no language input) and after
Stage 2 (with task instructions) across both simulation and
real-world settings.

*Stage 1: Emergent Physical Saliency.* Without any text
prompt, the pretrained model's attention (middle column)
automatically concentrates on the robot gripper and nearby
objects—the carrot in simulation, the pumpkin in the real
world. Background elements (wood texture, floor) are suppressed. This behavior emerges directly from the inverse
dynamics objective: predicting actions from observation
pairs $(o_t, o_{t+1})$ forces the encoder to track end-effector
kinematics and object interactions. The result is an implicit *affordance map* that identifies manipulable entities
without task specification.

*Stage 2: Semantic Grounding and Execution Focus.* Upon
receiving a language instruction (right column), we observe
a distinct shift in attention dynamics: the heatmap becomes
**intensely concentrated on the robotic gripper**. Unlike
Stage 1, where attention is distributed across multiple potential affordances, the language prompt acts as a strictly
constraining filter. It effectively "prunes" away irrelevant
physical possibilities (distractors), forcing the model to lock
its visual processing resources onto the *agent of action* (the
gripper) to ensure precise motor execution. This confirms
that while Stage 1 builds a broad space of physical possibilities, Stage 2 collapses this space into a singular, focused
point of execution.

*Cross-Domain Transfer.* The bottom row demonstrates robustness under domain shift. Despite deployment in a real-world environment with novel backgrounds and lighting, the pretrained attention maps maintain consistent focus on the gripper and interactive objects. This suggests that the learned physical representations capture domain-invariant structure rather than overfitting to simulation textures.

### 4.5. Error Analysis

To provide a comprehensive understanding of our TAP method, we systematically analyzed the failure cases encountered during our real-world WidowX experiments. Grounded in our Decomposition Hypothesis, we categorize the failures into two primary modes: Execution Failures (deficits in "how to move") and Semantic Failures (deficits in "what to do").

**Execution and Dynamics Failures (Approx. 25% of failures):** These errors occur when the policy correctly identifies the target object and attempts the right sub-task, but fails during fine-grained physical contact. Common manifestations include the end-effector slipping off the object, millimetric pre-grasp misalignment, or depth ambiguity caused by singular camera viewpoints. In the task 'push the pumpkin to the left', failures may also occur due to While our task-agnostic pretraining significantly mitigates these issues compared to standard BC, purely reactive VLA models still struggle with complex, out-of-distribution 3D spatial reasoning under extreme visual shifts.

**Semantic and Reasoning Failures (Approx. 75% of failures):** These errors are characterized by flawless physical execution directed at the wrong semantic goal. For instance, in the presence of visual distractors, the robot might execute a perfectly smooth grasp on a distractor object rather than the target instruction. Alternatively, in longer horizon sequences, the model occasionally experiences "freezing" or repetitive looping, losing track of the overarching linguistic instruction. This suggests that while the lower-level execution capabilities are robust, the implicit reasoning capacity of a singular, reactive VLA model remains a bottleneck.

## 5. Limitations and Future Work

While TAP validates the Decomposition Hypothesis by effectively extracting physical affordances from task-agnostic data, our framework still has several primary limitations:

**The Complexity Bound of Task-Agnostic Data:** Our pretraining relies on discarded trajectories and autonomous random play. While abundant, this data predominantly captures simple contact dynamics (e.g., pushing, basic grasping). Consequently, the learned physical priors currently still lack the structural depth required for highly dexterous, long-horizon tasks, such as tool use or articulated object

manipulation.

**The Static Legacy of VLM Backbones:** Current VLA architectures inherit backbones pretrained on static image-text pairs. Although TAP grounds these models physically via an Inverse Dynamics objective ($o_t \rightarrow a_t \rightarrow o_{t+1}$), it remains a reactive mapping. The architecture lacks a true Embodied World Model capable of forward prediction (simulating future states conditioned on actions), which fundamentally limits its capacity for complex spatial causality.

To overcome these bottlenecks, future work should transition from passive data collection to *active, curiosity-driven exploration*, enabling robots to autonomously seek out high-entropy, complex physical interactions. Furthermore, integrating a Forward Dynamics objective alongside TAP presents a highly promising direction. By jointly predicting actions (Inverse) and hallucinating future visual states (Forward), future architectures can evolve static VLM backbones into predictive physical simulators, bridging the gap between reactive control and true Embodied World Models.

## 6. Conclusion

We introduced **Task-Agnostic Pretraining (TAP)**, a two-stage framework that decouples the learning of physical affordances from semantic task understanding in Vision-Language-Action models. Our key insight—the *Decomposition Hypothesis*—posits that "how to move" can be learned entirely from cheap, unlabeled interaction data, reserving expensive expert demonstrations for teaching "what to do."

Our experiments yield three principal findings. First, **task-agnostic data is surprisingly effective**: by pretraining on off-task trajectories or autonomous random play via an Inverse Dynamics objective, TAP achieves a 10% absolute gain over standard Behavior Cloning and matches models trained on 1M+ expert trajectories—using orders of magnitude less labeled data. Second, **physical grounding transfers across tasks**: the boost in partial success from 31.8% to 45.8% confirms that self-supervised pretraining instills generalizable motor competencies rather than task-specific behaviors. Third, **self-exploration breeds robustness**: in real-world experiments, TAP retains 15–25% success under camera perturbations that cause catastrophic failure (0%) in internet-scale baselines, demonstrating that diverse physical experience yields domain-invariant representations.

These results challenge the prevailing assumption that scaling expert data is the only path to capable embodied agents. Instead, we show that active, task-agnostic interaction—akin to infant-like motor babbling—provides a complementary and cost-effective foundation for robot learning.

## Acknowledgements

This work was supported by the National Natural Science Foundation of China (No. 62521004).

Additionally, we would like to express our sincere gratitude to Chunbiao Feng and Hongbo Tang for their invaluable assistance with the hardware setup and control implementations.

## Impact Statement

This paper presents work where the goal is to advance the fields of machine learning and robotics. There are many potential societal consequences of our work, none of which we feel must be specifically highlighted here.

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

# A. Details of Autonomous Random Play Data Collection

To ensure that autonomous exploration yields safe, contact-rich physical interactions rather than redundant free-space motions, we implement a two-phase constrained procedural generation framework.

**Phase 1: Safe Workspace Initialization.**  The first step for autonomous play is to establish a sufficient safe spatial prior. An operator first teleoperates the robot without specific task instructions to perform regular movements and densely cover the reachable workspace. After filtering out kinematically unsafe or out-of-bound poses, we apply Voxel Grid Downsampling (e.g., with a leaf size of $5\text{cm}^3$) to the retained poses. This mitigates spatial density bias and yields a uniform, safe, and discrete pose library $\mathcal{P}_{safe}$.

**Phase 2: Constrained Trajectory Generation and Execution.**  To procedurally generate trajectories, we stochastically sample sequences of waypoints from $\mathcal{P}_{safe}$ under a minimum distance constraint. To guarantee contact-rich interactions (e.g., pushing, sliding) and prevent the end-effector from hovering, we apply a contact-forcing heuristic: any generated sequence that remains above a specified elevation threshold ($z_{thresh}$) for more than $c_{max}$ consecutive steps is geometrically adjusted to force a descent, ensuring frequent engagement with the tabletop.

The modified waypoints are then connected via cosine interpolation, and boundary-aware Gaussian noise is injected to enhance trajectory diversity. During execution, the robot continuously performs these trajectories, with human intervention strictly limited to periodic resets (e.g., every 30 minutes) to add, remove or shuffle interactive objects.

During data collection, raw trajectories are typically collected at high control frequencies (e.g., 25Hz), yielding minimal visual displacement between adjacent frames. At such rates, the inverse dynamics task becomes ill-posed: the action signal is dominated by sensor noise rather than meaningful motion. To address this, we downsample all training data to **5Hz**, ensuring that (1) the visual change between $o_t$ and $o_{t+1}$ is perceptible and causally attributable to the executed action, and (2) the model learns semantically meaningful action primitives (e.g., "approach," "grasp," "lift") rather than micro-adjustments.

# B. Training and Evaluation Details

## B.1. Model Architecture

We instantiate our framework using **Qwen2.5-VL** (3B parameters) as the VLM backbone. The visual encoder is a ViT-based **SigLIP** (400M parameters), which processes input images at $224\times224$ resolution and produces a sequence of visual tokens. The action head is a lightweight 2-layer MLP that projects the VLM's last hidden state to the 7-dimensional action space. During Stage 1, we freeze the visual encoder and train only the VLM backbone and action head; in Stage 2, all parameters are jointly finetuned.

## B.2. Action Representation

We adopt a **delta-pose end-effector action space**, where $a_t \in \mathbb{R}^7$ encodes the relative position change $(\Delta x, \Delta y, \Delta z)$, orientation change (represented as a 3D axis-angle vector), and a scalar gripper command. Predicting relative motion rather than absolute poses enables the model to learn local interaction dynamics that are invariant to global workspace coordinates—a property critical for transferring physical priors across different robot configurations.

## B.3. Training Details.

We train our model for $100,000$ steps on a single node equipped with 8 NVIDIA H100 GPUs. The training is implemented using the Hugging Face Accelerate library to ensure efficient distributed execution. We use a global batch size of 128 (16 per GPU). The model is optimized using the AdamW optimizer with a weight decay of $0.05$ and standard $\beta$ parameters ($\beta_1 = 0.9, \beta_2 = 0.999$). The learning rate is initialized at $5 \times 10^{-5}$ and follows a cosine decay schedule with a warmup ratio of $0.05$ (warming up for the first $5,000$ steps). To stabilize training, we apply global gradient clipping with a max norm of $1.0$. For computational efficiency and numerical stability, all training runs are conducted in `bfloat16` precision.

## B.4. Evaluation Details

**Simpler Evaluation**  For Simpler Evaluation, we follow the standard SIMPLER protocol, reporting success rates averaged over 50 episodes per checkpoint. All comparing models are trained with the same amount of Bridge data with same training

settings.

**Real World Evaluation Details.** To further validate the effectiveness and robustness of the **TAP** framework, we conduct real-world experiments on two manipulation tasks: *"Put Carrot on Plate"* and *"Push Pumpkin to Left"*. Inspired by the generalization aspects introduced by previous works(Shi et al., 2025; Fei et al., 2025), we evaluate the policy across five distinct scenarios designed to test generalization: In-Domain setups, Initial State Perturbations, Visual Distractors, Background Texture Shifts, and Viewpoint Variations.

For each task, we conduct 20 evaluation trials per scenario. The detailed protocols for each scenario are listed as follows:

- **Standard Setup (In-Domain):** The experimental environment strictly replicates the training setting, ensuring no variations in object positions, lighting, or background. This serves as the baseline for assessing the upper bound of model performance.

- **Initial State Perturbation:** To evaluate the model's robustness to initial conditions, we apply random spatial perturbations to the robot's home pose (starting position). This tests the policy's ability to recover and complete the task from unseen starting configurations.

- **Visual Distractors:** We introduce unseen objects to test robustness against visual clutter. For this setting, we create five distinct combinations, each containing three to five random distractor objects placed on the table. Crucially, these distractors are positioned without obstructing the manipulation trajectory to ensure the task remains physically feasible. Each combination is tested over 4 trials (totaling 20 trials), and the average success rate is reported.

- **Background Texture Shift:** We evaluate the model's invariance to background changes by placing four different tablecloths (layers) with varying textures on the table, while maintaining the original object arrangements. Each background texture is evaluated over 5 trials (totaling 20 trials).

- **Viewpoint Variation:** To assess robustness against camera calibration noise, we apply minor shifts to the extrinsic parameters (e.g., angle, pitch) of the third-person view camera. These perturbations generate four distinct camera viewpoints that differ slightly from the training view while preserving the main visual semantics. Each viewpoint configuration is tested over 5 trials (totaling 20 trials).

Finally, the overall average success rate is calculated as the arithmetic mean across all five evaluation scenarios.

## C. Data Efficiency and Computational Cost Analysis

A core motivation of our work is to democratize generalist robot learning by reducing the dependency on massive, curated expert datasets and prohibiting computational budgets. In this section, we provide a detailed comparison of data requirements and training costs between our proposed TAP framework and current state-of-the-art Large-Scale VLA baselines.

### C.1. Data Comparison: Expert vs. Task-Agnostic

As illustrated in Table 4, standard VLA models (e.g., OpenVLA, Octo, NORA) rely heavily on the Open X-Embodiment (OXE) dataset, which aggregates over 2 million expert trajectories across varying embodiments. While effective, curating and standardizing such datasets requires immense human effort.

In contrast, our TAP framework minimizes the reliance on expert data.

- **Stage 1 (Pretraining):** We utilize purely autonomous, task-agnostic interaction data (e.g., random exploration or play). This data is "free" in terms of human labeling cost.

- **Stage 2 (Finetuning):** We achieve competitive performance using only a fraction of the expert demonstrations (e.g., 200 trajectories in real-world experiments) compared to the millions seen by baselines.

Quantitatively, our approach reduces the demand for expert data by several orders of magnitude while maintaining comparable manipulation proficiency.

*Table 4.* **Comparison of Data Scale and Computational Resources.** We compare our TAP against leading VLA baselines. *Expert Data* refers to human-teleoperated or curated demonstrations. *Cheap Data* refers to autonomous, unlabeled interactions (e.g., self-play). Note that our method achieves competitive results using significantly less expert data and manageable compute resources compared to models trained on the massive Open X-Embodiment (OXE) dataset.

| Model | Pretraining Dataset | Expert Data Scale | Task-Agnostic (Cheap) Data | Compute Infrastructure | Training Objective |
|---|---|---|---|---|---|
| *Baselines* | | | | | |
| RT-1-X (Collaboration et al., 2023) | Open X-Embodiment | ∼1M Trajectories | None | TPU v4 Pods | BC (Categorical) |
| Octo (Team et al., 2024) | Open X-Embodiment | ∼800k Trajectories | None | TPU v4-128 | Diffusion |
| OpenVLA (Kim et al., 2024) | Open X-Embodiment | ∼970k Trajectories | None | 64 × A100 | Llama-2 Finetuning |
| NORA (Hung et al., 2025) | Open X-Embodiment | ∼970k Trajectories | None | - | VLA Finetuning |
| *Ours* | | | | | |
| **TAP** | **Self-Generated Play** | **< 1k Trajectories** | **∼100k Steps** | **8 × H100** | **Inverse Dynamics + BC** |

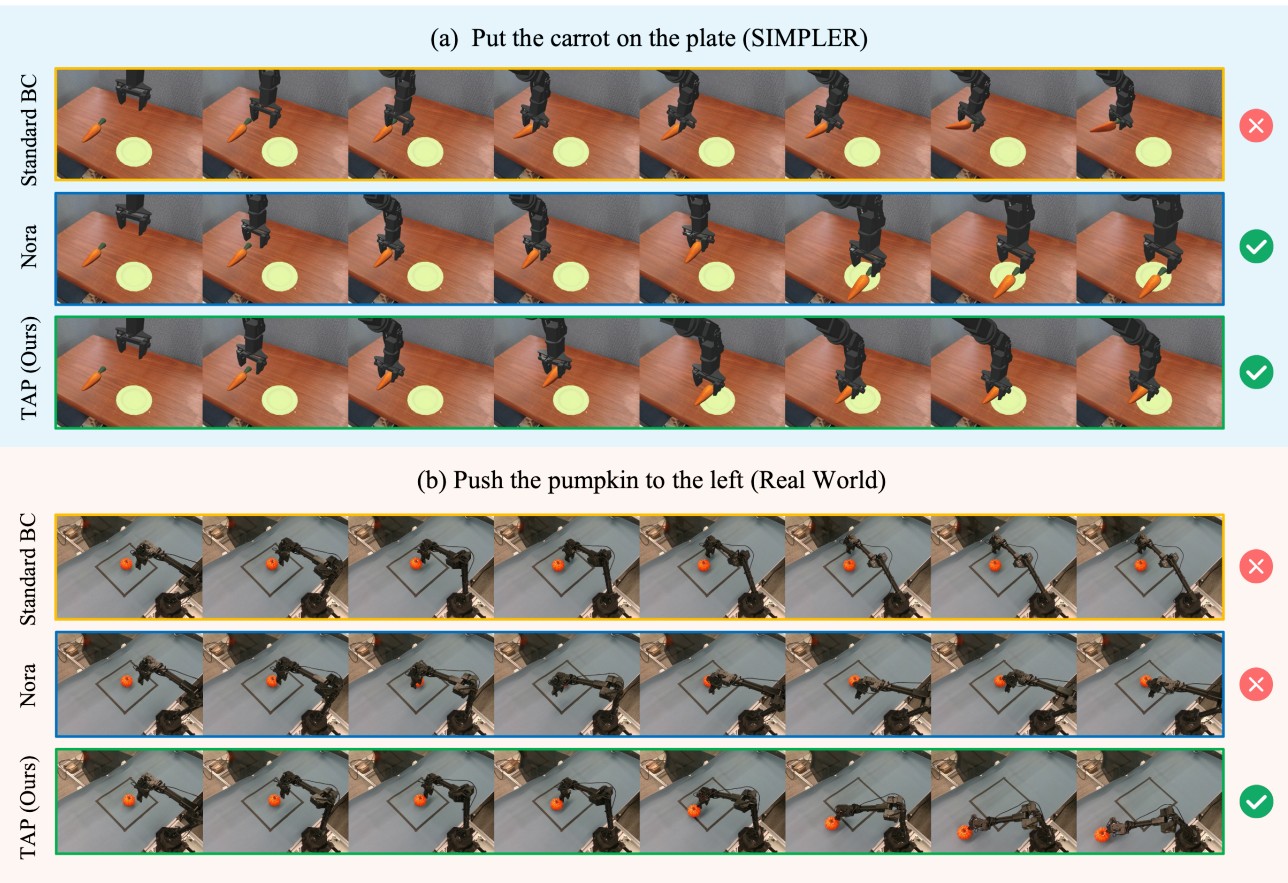

*Figure 6.* **Qualitative Comparison in Simulation and Real-World Environments.** (a) On the SIMPLER "Put the carrot on the plate" task, Standard BC fails to execute the final grasp, while Nora and TAP succeed. (b) On the real-world "Push the pumpkin to the left" task under an unseen background, Standard BC exhibits poor contact. The Nora baseline suffers from a visual grounding failure; it attempts a leftward push but misjudges the pumpkin's location due to the novel texture, pushing empty space. TAP accurately isolates the object despite the background shift and successfully completes the task.

## C.2. Computational Budget

Training foundation models like OpenVLA typically necessitates industrial-scale infrastructure (e.g., TPU v4 Pods or clusters of A100s) running for weeks. Our method is designed for academic-scale resources. As detailed in Table 4, our pretraining and finetuning can be completed on a single node with 8×H100 GPUs within a reasonable timeframe (approx. 24 GPU hours), making the reproduction and iteration of VLA policies significantly more accessible.

# D. Qualitative Analysis and Case Studies

To better understand the mechanisms driving the quantitative improvements, we conduct a qualitative analysis comparing our TAP framework against Standard BC and Nora baselines across both simulation and real-world environments, as visualized in Figure 6.

## D.1. Simulation: Unlocking Task-Irrelevant Data in SIMPLER

A core claim of our Decomposition Hypothesis is that the physical affordances required for manipulation can be extracted from task-agnostic data. Figure 6(a) illustrates a comparison on the SIMPLER "Put the carrot on the plate" task.

When expert data is scarce, the Standard BC model struggles to ground the linguistic instruction in precise 3D geometry. As shown in the top row, the BC policy navigates to the general vicinity of the carrot but halts, failing to execute the fine-grained contact dynamics required for a successful grasp. Both the Nora baseline and our TAP demonstrate robust physical execution, successfully grasping the carrot and placing it on the plate. For TAP, this confirms that "how to move" (precise pre-grasp alignment and contact) transfers effectively even when the model is pretrained on discarded, task-irrelevant trajectories via Inverse Dynamics.

## D.2. Real-World: Robustness to Unseen Background Shifts

In our real-world WidowX 250 experiments, the distinction between robust physical grounding and brittle visual matching becomes starkly apparent. Figure 6(b) visualizes the "Push the pumpkin to the left" task evaluated on an unseen background texture, a scenario designed to test generalization beyond the training distribution.

The Standard BC model struggles with basic execution, extending the arm but failing to make proper, sustained contact with the object. Interestingly, the Nora baseline exhibits a severe visual grounding failure induced by the out-of-distribution background. While it attempts the semantically correct trajectory (moving its end-effector towards the left), it misjudges the precise 3D spatial coordinates of the pumpkin against the novel table texture. Consequently, it completely misses the object, pushing empty space to the left of the pumpkin. In contrast, TAP successfully leverages its robust, task-agnostic physical priors to accurately isolate the manipulable object from the novel background. It makes solid contact and deliberately pushes the pumpkin to the correct side, demonstrating superior domain-invariant spatial awareness.

# E. Full Experimental Results

Due to space limitations in the main text, we present the comprehensive breakdown of our experimental results in Table 5, including performance metrics across all individual evaluation checkpoints and sub-tasks.

*Table 5.* **Comprehensive Scaling Results (Origin Method).** Detailed breakdown of success rates across all combinations of Stage 1 (pretraining) and Stage 2 (finetuning) steps. Steps are reported in thousands (k).

| Training Steps (k) | | Spoon on cloth | | Carrot on plate | | Stack Blocks | | Eggplant in Basket | | | | |
| Stage 1 | Stage 2 | Part. | Ent. | Part. | Ent. | Part. | Ent. | Part. | Ent. | Avg-P | Avg-E | Avg-All |
|---|---|---|---|---|---|---|---|---|---|---|---|---|
| 20 | 10 | 8.3% | 0.0% | 12.5% | 0.0% | 25.0% | 0.0% | 0.0% | 0.0% | 11.45% | 0.00% | 5.73% |
| 20 | 20 | 16.7% | 4.2% | 37.5% | 0.0% | 29.2% | 0.0% | 0.0% | 0.0% | 20.85% | 1.05% | 10.95% |
| 20 | 30 | 16.7% | 4.2% | 25.0% | 0.0% | 29.2% | 0.0% | 0.0% | 0.0% | 17.72% | 1.05% | 9.39% |
| 20 | 40 | 33.3% | 16.7% | 45.8% | 12.5% | 75.0% | 4.2% | 0.0% | 0.0% | 38.52% | 8.35% | 23.44% |
| 20 | 50 | 33.3% | 25.0% | 33.3% | 0.0% | 62.5% | 16.7% | 0.0% | 0.0% | 32.27% | 10.43% | 21.35% |
| 20 | 60 | 16.7% | 12.5% | 25.0% | 0.0% | 62.5% | 16.7% | 0.0% | 0.0% | 26.05% | 7.30% | 16.68% |
| 20 | 70 | 20.8% | 8.3% | 41.7% | 4.2% | 50.0% | 8.3% | 0.0% | 0.0% | 28.12% | 5.20% | 16.66% |
| 20 | 80 | 8.3% | 0.0% | 33.3% | 0.0% | 58.3% | 12.5% | 4.2% | 0.0% | 26.02% | 3.12% | 14.57% |
| 20 | 90 | 25.0% | 4.2% | 29.2% | 4.2% | 66.7% | 20.8% | 0.0% | 0.0% | 30.23% | 7.30% | 18.76% |
| 20 | 100 | 25.0% | 8.3% | 33.3% | 4.2% | 62.5% | 16.7% | 0.0% | 0.0% | 30.20% | 7.30% | 18.75% |
| 30 | 10 | 0.0% | 0.0% | 0.0% | 0.0% | 20.8% | 0.0% | 0.0% | 0.0% | 5.20% | 0.00% | 2.60% |
| 30 | 20 | 8.3% | 0.0% | 25.0% | 0.0% | 41.7% | 4.2% | 0.0% | 0.0% | 18.75% | 1.05% | 9.90% |
| 30 | 30 | 29.2% | 12.5% | 25.0% | 4.2% | 33.3% | 0.0% | 4.2% | 0.0% | 22.93% | 4.17% | 13.55% |
| 30 | 40 | 37.5% | 16.7% | 29.2% | 0.0% | 50.0% | 8.3% | 0.0% | 0.0% | 29.18% | 6.25% | 17.71% |
| 30 | 50 | 29.2% | 16.7% | 33.3% | 0.0% | 45.8% | 4.2% | 0.0% | 0.0% | 27.07% | 5.23% | 16.15% |

**Table 5 – continued from previous page**

| Training Steps (k) | | Spoon on cloth | | Carrot on plate | | Stack Blocks | | Eggplant in Basket | | | | |
|---|---|---|---|---|---|---|---|---|---|---|---|---|
| Stage1 | Stage2 | Part. | Ent. | Part. | Ent. | Part. | Ent. | Part. | Ent. | Avg-P | Avg-E | Avg-All |
| 30 | 60 | 29.2% | 16.7% | 41.7% | 8.3% | 41.7% | 0.0% | 4.2% | 4.2% | 29.20% | 7.30% | 18.25% |
| 30 | 70 | 41.7% | 16.7% | 37.5% | 8.3% | 37.5% | 0.0% | 8.3% | 8.3% | 31.25% | 8.33% | 19.79% |
| 30 | 80 | 25.0% | 20.8% | 37.5% | 4.2% | 45.8% | 4.2% | 4.2% | 4.2% | 28.12% | 8.35% | 18.24% |
| 30 | 90 | 25.0% | 12.5% | 25.0% | 0.0% | 41.7% | 8.3% | 0.0% | 0.0% | 22.93% | 5.20% | 14.06% |
| 30 | 100 | 20.8% | 8.3% | 33.3% | 0.0% | 41.7% | 4.2% | 0.0% | 0.0% | 23.95% | 3.12% | 13.54% |
| 40 | 10 | 0.0% | 0.0% | 0.0% | 0.0% | 16.7% | 0.0% | 0.0% | 0.0% | 4.17% | 0.00% | 2.09% |
| 40 | 20 | 25.0% | 12.5% | 12.5% | 0.0% | 37.5% | 0.0% | 12.5% | 0.0% | 21.88% | 3.12% | 12.50% |
| 40 | 30 | 16.7% | 12.5% | 29.2% | 12.5% | 58.3% | 8.3% | 0.0% | 0.0% | 26.05% | 8.33% | 17.19% |
| 40 | 40 | 37.5% | 29.2% | 33.3% | 4.2% | 45.8% | 4.2% | 8.3% | 0.0% | 31.23% | 9.40% | 20.31% |
| 40 | 50 | 33.3% | 16.7% | 37.5% | 0.0% | 62.5% | 8.3% | 0.0% | 0.0% | 33.32% | 6.25% | 19.79% |
| 40 | 60 | 25.0% | 16.7% | 29.2% | 4.2% | 58.3% | 8.3% | 4.2% | 0.0% | 29.18% | 7.30% | 18.24% |
| 40 | 70 | 33.3% | 20.8% | 16.7% | 4.2% | 50.0% | 8.3% | 4.2% | 0.0% | 26.05% | 8.33% | 17.19% |
| 40 | 80 | 45.8% | 29.2% | 33.3% | 0.0% | 50.0% | 4.2% | 0.0% | 0.0% | 32.27% | 8.35% | 20.31% |
| 40 | 90 | 50.0% | 37.5% | 37.5% | 8.3% | 58.3% | 4.2% | 0.0% | 0.0% | 36.45% | 12.50% | 24.47% |
| 40 | 100 | 50.0% | 37.5% | 12.5% | 0.0% | 50.0% | 8.3% | 4.2% | 0.0% | 29.18% | 11.45% | 20.31% |
| 50 | 10 | 0.0% | 0.0% | 8.3% | 0.0% | 25.0% | 0.0% | 0.0% | 0.0% | 8.33% | 0.00% | 4.16% |
| 50 | 20 | 12.5% | 4.2% | 8.3% | 0.0% | 33.3% | 4.2% | 0.0% | 0.0% | 13.53% | 2.10% | 7.81% |
| 50 | 30 | 33.3% | 16.7% | 29.2% | 0.0% | 37.5% | 4.2% | 0.0% | 0.0% | 25.00% | 5.23% | 15.11% |
| 50 | 40 | 33.3% | 16.7% | 37.5% | 4.2% | 54.2% | 0.0% | 4.2% | 4.2% | 32.30% | 6.28% | 19.29% |
| 50 | 50 | 37.5% | 25.0% | 33.3% | 8.3% | 45.8% | 12.5% | 20.8% | 8.3% | 34.35% | 13.53% | 23.94% |
| 50 | 60 | 37.5% | 20.8% | 37.5% | 0.0% | 50.0% | 4.2% | 4.2% | 4.2% | 32.30% | 7.30% | 19.80% |
| 50 | 70 | 29.2% | 16.7% | 37.5% | 4.2% | 41.7% | 0.0% | 25.0% | 12.5% | 33.35% | 8.35% | 20.85% |
| 50 | 80 | 25.0% | 16.7% | 33.3% | 8.3% | 50.0% | 0.0% | 29.2% | 8.3% | 34.38% | 8.33% | 21.35% |
| 50 | 90 | 29.2% | 25.0% | 37.5% | 8.3% | 45.8% | 0.0% | 20.8% | 4.2% | 33.32% | 9.38% | 21.35% |
| 50 | 100 | 29.2% | 20.8% | 37.5% | 0.0% | 54.2% | 4.2% | 16.7% | 12.5% | 34.40% | 9.38% | 21.89% |
| 60 | 10 | 0.0% | 0.0% | 0.0% | 0.0% | 25.0% | 0.0% | 4.2% | 0.0% | 7.30% | 0.00% | 3.65% |
| 60 | 20 | 25.0% | 12.5% | 33.3% | 0.0% | 45.8% | 4.2% | 0.0% | 0.0% | 26.02% | 4.17% | 15.10% |
| 60 | 30 | 20.8% | 0.0% | 8.3% | 0.0% | 37.5% | 0.0% | 8.3% | 0.0% | 18.73% | 0.00% | 9.36% |
| 60 | 40 | 29.2% | 25.0% | 33.3% | 4.2% | 58.3% | 0.0% | 8.3% | 0.0% | 32.27% | 7.30% | 19.79% |
| 60 | 50 | 20.8% | 8.3% | 37.5% | 0.0% | 50.0% | 16.7% | 0.0% | 0.0% | 27.07% | 6.25% | 16.66% |
| 60 | 60 | 29.2% | 16.7% | 29.2% | 4.2% | 45.8% | 4.2% | 8.3% | 0.0% | 28.12% | 6.28% | 17.20% |
| 60 | 70 | 25.0% | 12.5% | 37.5% | 8.3% | 58.3% | 12.5% | 4.2% | 0.0% | 31.25% | 8.33% | 19.79% |
| 60 | 80 | 41.7% | 20.8% | 20.8% | 4.2% | 66.7% | 12.5% | 8.3% | 0.0% | 34.38% | 9.38% | 21.88% |
| 60 | 90 | 41.7% | 20.8% | 29.2% | 8.3% | 58.3% | 12.5% | 0.0% | 0.0% | 32.30% | 10.40% | 21.35% |
| 60 | 100 | 29.2% | 12.5% | 33.3% | 4.2% | 62.5% | 8.3% | 4.2% | 0.0% | 32.30% | 6.25% | 19.28% |
| 70 | 10 | 20.8% | 0.0% | 20.8% | 0.0% | 25.0% | 0.0% | 4.2% | 0.0% | 17.70% | 0.00% | 8.85% |
| 70 | 20 | 16.7% | 4.2% | 29.2% | 0.0% | 45.8% | 0.0% | 4.2% | 0.0% | 23.97% | 1.05% | 12.51% |
| 70 | 30 | 12.5% | 12.5% | 29.2% | 0.0% | 45.8% | 0.0% | 4.2% | 0.0% | 22.93% | 3.12% | 13.03% |
| 70 | 40 | 54.2% | 25.0% | 41.7% | 0.0% | 79.2% | 16.7% | 8.3% | 0.0% | 45.85% | 10.43% | 28.14% |
| 70 | 50 | 33.3% | 16.7% | 41.7% | 4.2% | 83.3% | 16.7% | 12.5% | 0.0% | 42.70% | 9.40% | 26.05% |
| 70 | 60 | 41.7% | 37.5% | 50.0% | 12.5% | 62.5% | 4.2% | 8.3% | 0.0% | 40.62% | 13.55% | 27.09% |
| 70 | 70 | 41.7% | 33.3% | 50.0% | 16.7% | 83.3% | 12.5% | 4.2% | 0.0% | 44.80% | 15.62% | 30.21% |
| 70 | 80 | 50.0% | 29.2% | 45.8% | 8.3% | 70.8% | 8.3% | 4.2% | 0.0% | 42.70% | 11.45% | 27.07% |
| 70 | 90 | 41.7% | 25.0% | 41.7% | 8.3% | 70.8% | 16.7% | 12.5% | 4.2% | 41.67% | 13.55% | 27.61% |
| 70 | 100 | 50.0% | 41.7% | 45.8% | 8.3% | 70.8% | 8.3% | 0.0% | 0.0% | 41.65% | 14.57% | 28.11% |
| 80 | 10 | 0.0% | 0.0% | 12.5% | 0.0% | 25.0% | 0.0% | 0.0% | 0.0% | 9.38% | 0.00% | 4.69% |
| 80 | 20 | 12.5% | 0.0% | 29.2% | 0.0% | 41.7% | 0.0% | 0.0% | 0.0% | 20.85% | 0.00% | 10.42% |
| 80 | 30 | 33.3% | 16.7% | 20.8% | 0.0% | 33.3% | 0.0% | 4.2% | 0.0% | 22.90% | 4.17% | 13.54% |
| 80 | 40 | 29.2% | 12.5% | 37.5% | 4.2% | 41.7% | 0.0% | 0.0% | 0.0% | 27.10% | 4.17% | 15.64% |
| 80 | 50 | 33.3% | 12.5% | 33.3% | 4.2% | 41.7% | 4.2% | 0.0% | 0.0% | 27.07% | 5.23% | 16.15% |
| 80 | 60 | 4.2% | 0.0% | 37.5% | 4.2% | 45.8% | 8.3% | 12.5% | 8.3% | 25.00% | 5.20% | 15.10% |
| 80 | 70 | 41.7% | 20.8% | 41.7% | 0.0% | 66.7% | 4.2% | 12.5% | 0.0% | 40.65% | 6.25% | 23.45% |
| 80 | 80 | 33.3% | 25.0% | 33.3% | 12.5% | 54.2% | 8.3% | 4.2% | 4.2% | 31.25% | 12.50% | 21.88% |
| 80 | 90 | 20.8% | 8.3% | 37.5% | 12.5% | 62.5% | 8.3% | 0.0% | 0.0% | 30.20% | 7.28% | 18.74% |
| 80 | 100 | 41.7% | 33.3% | 29.2% | 4.2% | 54.2% | 4.2% | 4.2% | 4.2% | 32.32% | 11.47% | 21.90% |
| 90 | 10 | 0.0% | 0.0% | 0.0% | 0.0% | 16.7% | 0.0% | 0.0% | 0.0% | 4.17% | 0.00% | 2.09% |

**Table 5 – continued from previous page**

| Training Steps (k) | | Spoon on cloth | | Carrot on plate | | Stack Blocks | | Eggplant in Basket | | Avg-P | Avg-E | Avg-All |
|---|---|---|---|---|---|---|---|---|---|---|---|---|
| Stage1 | Stage2 | Part. | Ent. | Part. | Ent. | Part. | Ent. | Part. | Ent. | | | |
| 90 | 20 | 0.0% | 0.0% | 16.7% | 0.0% | 25.0% | 4.2% | 0.0% | 0.0% | 10.43% | 1.05% | 5.74% |
| 90 | 30 | 25.0% | 8.3% | 29.2% | 0.0% | 54.2% | 4.2% | 4.2% | 0.0% | 28.15% | 3.12% | 15.64% |
| 90 | 40 | 41.7% | 20.8% | 45.8% | 4.2% | 58.3% | 0.0% | 0.0% | 0.0% | 36.45% | 6.25% | 21.35% |
| 90 | 50 | 20.8% | 12.5% | 45.8% | 0.0% | 62.5% | 8.3% | 4.2% | 0.0% | 33.32% | 5.20% | 19.26% |
| 90 | 60 | 37.5% | 16.7% | 58.3% | 4.2% | 62.5% | 4.2% | 0.0% | 0.0% | 39.57% | 6.28% | 22.93% |
| 90 | 70 | 37.5% | 25.0% | 41.7% | 0.0% | 58.3% | 4.2% | 0.0% | 0.0% | 34.38% | 7.30% | 20.84% |
| 90 | 80 | 37.5% | 16.7% | 41.7% | 0.0% | 58.3% | 20.8% | 0.0% | 0.0% | 34.38% | 9.38% | 21.88% |
| 90 | 90 | 37.5% | 25.0% | 37.5% | 0.0% | 54.2% | 8.3% | 0.0% | 0.0% | 32.30% | 8.33% | 20.31% |
| 90 | 100 | 29.2% | 16.7% | 33.3% | 0.0% | 50.0% | 12.5% | 4.2% | 0.0% | 29.18% | 7.30% | 18.24% |
| 100 | 10 | 0.0% | 0.0% | 25.0% | 0.0% | 16.7% | 0.0% | 0.0% | 0.0% | 10.43% | 0.00% | 5.21% |
| 100 | 20 | 16.7% | 4.2% | 12.5% | 0.0% | 37.5% | 8.3% | 0.0% | 0.0% | 16.68% | 3.12% | 9.90% |
| 100 | 30 | 50.0% | 16.7% | 50.0% | 0.0% | 70.8% | 12.5% | 8.3% | 0.0% | 44.77% | 7.30% | 26.04% |
| 100 | 40 | 41.70% | 20.80% | 45.80% | 4.20% | 58.30% | 0.00% | 0.00% | 0.00% | 36.45% | 6.25% | 21.35% |
| 100 | 50 | 50.0% | 29.2% | 50.0% | 4.2% | 70.8% | 8.3% | 12.5% | 4.2% | 45.82% | 11.47% | 28.65% |
| 100 | 60 | 50.00% | 25.00% | 41.70% | 8.30% | 58.30% | 16.70% | 8.30% | 0.00% | 39.57% | 12.50% | 26.04% |
| 100 | 70 | 62.5% | 29.2% | 45.8% | 12.5% | 83.3% | 12.5% | 4.2% | 4.2% | 48.95% | 14.60% | 31.77% |
| 100 | 80 | 45.8% | 25.0% | 25.0% | 4.2% | 83.3% | 8.3% | 20.8% | 12.5% | 43.72% | 12.50% | 28.11% |
| 100 | 90 | 41.7% | 29.2% | 41.7% | 8.3% | 66.7% | 0.0% | 16.7% | 16.7% | 41.70% | 13.55% | 27.62% |
| 100 | 100 | 66.70% | 58.30% | 50.00% | 0.00% | 58.30% | 16.70% | 8.30% | 8.30% | 45.82% | 20.82% | 33.32% |

