# OpenReview forum: "Learning to Move Before Learning to Do: Task-Agnostic pretraining for VLAs"
_ICML.cc/2026/Conference — ICML 2026 regular_

### Official Review · Reviewer_6cXk · 2026-03-12

**Soundness:** 2
**Presentation:** 1
**Significance:** 3
**Originality:** 3
**Overall Recommendation:** 4
**Confidence:** 4

**Summary:**

The paper proposed a two-stage framework, which pre-trains on task-agnostic data and then post-trains with language instructions. This two-stage training pipeline achieves the training efficiency and stronger performance in the provided experiments. The proposed method is evaluated in a simulation benchmark and in the real world.

**Compliance With Llm Reviewing Policy:**

Affirmed.

**Final Justification:**

I have raised my score as the rebuttal addressed my main concerns and now I think the paper indeed brings some new insights.

My evaluation focuses more on the new insight it brings. I agree that there are some flaws regarding the soundness of the work. The clarity is not good but I think the author could easily improve this in the revised version.

**Key Questions For Authors:**

1. How much did the existing dataset and the autonomous exploration dataset **respectively** contribute to the final performance?

I'm happy to raise my score if the author can show that the task-agnostic data is easy to collect automatically (without relying on existing annotated datasets) and contributes to the final performance.

**Limitations:**

I suggest that the author discuss the limitations of automatic data collection. How to set the automatic programme? Can the robot perform all the manipulations automatically (including grasping and interaction with other objects)?

**Strengths And Weaknesses:**

Strength:
1. **Good originality**. The proposed work offers novel insights into removing language from robotic pre-training. It demonstrates that a Vision-Action (VA) paradigm can also be highly effective, suggesting that the inclusion of language (as in VLA models) may not be a strict prerequisite for learning robust robotic representations.


Major Weakness:
1. **Lacking soundness in the task-agnostic pre-training.** While the model claims to leverage “task-agnostic” data for scaling, it still relies heavily on curated robotic demonstrations where the task labels have simply been stripped away. These so-called "unlabelled data" are just data without a task description, but it still requires the annotated robot actions to collect the robot demonstration. To truly demonstrate scalability, the approach should incorporate data that gets rid of robot action annotation, such as videos of human activities or non-robotic interactions, rather than relying on preprocessed, annotated robot trajectories.

2. **Unclear contribution of  "autonomous exploration data".** Although the proposed usage of "autonomous exploration data" seems quite inspiring, there's no ablation study to separate the contribution of these autonomous data. In addition, what is the annotation cost of defining the "randomized" end-effector commands is unclear. In Fig. 1, it seems that the robot is performing quite complex trajectories to grasp an object; this kind of trajectory cannot be generated "randomly". Overall, the concern is that the collection of task-agnostic data is still expensive, affecting the scalability of the method.

3. **Presentation needs improvement.** The manuscript is currently unpolished and shows a clear lack of proofreading. There are several quotation mark errors, paragraphing issues (Line 318),  and many improperly formatted references. Also, the layout is disorganized as the use of subsections, bullet points, and bold paragraphs is mixed up.

4. **Soundness in real-world experiments.** The real-world evaluation setting is too simple, with just two tasks and one baseline, failing to provide a comprehensive assessment.

Minor Weakness:
1. There is an inconsistency between Section 3.4 and Table 1. Section 3.4 defines actions as relative pose changes, while Table 1 shows behavior cloning, which might refer to joint angles.

---

> ### Author Rebuttal · Authors · 2026-03-31
>
> Dear Reviewer 6cXk,
>
> Thank you for your detailed and constructive review. We systematically address each concern below.
>
> **Response to Weakness 1:**
>
> We appreciate your critical perspective. In our framework, "task-agnostic" refers specifically to data **without semantic task descriptions or task-specific goals**, not necessarily data without action annotations.
>
> In simulation, Bridge data is stripped of task labels and instructions, demonstrating that discarded "off-task" data can be recycled at **zero annotation cost**. In real-world settings, our 30 hours of pretrain data consist entirely of **autonomous procedural play**, where the robot collects observation-action pairs without human guidance.
>
> Extending to action-free human videos is a valuable future direction. Our current focus maximizes utility of **cheap, scalable embodied data**—data robots can collect autonomously with minimal setup. We will clarify this scope in revised manuscript.
>
> **Response to Weakness 2:**
>
> As illustrated in Table 1 and 2, our main results table provides an implicit but comprehensive comparison:
>
> |Setting|Pre-trainingData|FinetuneData|Performance(Simpler)|
> |-|-|-|-|
> |Baseline|None|5k(sim)/200(real)|23.15%|
> |Nora|>1M teleop|same|20.06%|
> |TAP(Ours)|task-agnostic|same|33.32%|
>
> This demonstrates that our **task-agnostic pretraining alone—without any expert annotation—achieves comparable performance to models pretrained on 1M+ expert trajectories**.
>
> For annotation cost of "Randomized" Commands, we apologize for the ambiguity. Our autonomous play is **procedurally constrained**, not fully random.
>
> ---
> **Algorithm: Constrained Procedural Trajectory Generation**
>
> **Require:** Raw teleoperation joint poses $\mathcal{P}\_{raw}$, safety bounds $\mathcal{B}$, voxel size $v\_{size}$, maximum waypoint distance $d$, density function $\mathcal{p}\_{interact}$ ensures interaction, noise scale $\sigma$, Forward Kinematic function $FK$.
>
> **Phase 1: Safe Pose Library Initialization**
>
> $\mathcal{P}\_{safe}\leftarrow\\{ p \mid p \in\mathcal{P}\_{raw}\cap\mathcal{B}, P \in\mathbb{R}^{dof} \\}$
>
> $\mathcal{P}\leftarrow\text{VoxelDS}(\mathcal{P}\_{safe}, v\_{size}, FK)$ *(Note: VoxelGridDownsample in EEF)*
>
> **Phase 2: Autonomous Data Collection**
>
> $\mathcal{D}\_{play}\leftarrow\emptyset$
>
> **while** True **do**
>
> 　　$\mathcal{W}\sim\\{ (p\_{1}, \dots, p\_{\tau}) \mid p\_{i} \in\mathcal{P} \, \|FK(p\_{i}) - FK(p\_{i+1})\|\leq d\, \mathcal{E}\\lt \mathcal{p}_{interact}(FK\_{z}(p\_{i}))\ where\ \mathcal{E}\sim\mathcal{U}(0, M) \\}$ *(Reject Sampling)*
>
> 　　$\mathcal{W}\leftarrow CosineInterpolate(\mathcal{W}) \in\mathbb{R}^{n \times dof}$
>
> 　　$\mathcal{W}\sim\\{ (w\_{1}, \dots, w\_{n})\mid w\_{i} \sim\mathcal{N}(p\_{i}, \sigma) \, \forall p\_{i} \in\mathcal{W} \, w\_{i}\in\mathcal{B} \\}$
>
> 　　$\mathcal{D}\_{play}\leftarrow\mathcal{D}\_{play}\cup\\{ (\tau, o) \mid o \leftarrow\text{exec}(\tau), \tau\in\mathcal{W} \\}$
>
> 　　**if** human intervention triggered (e.g., $\Delta t \geq 30$ mins) **then**
>
> 　　　　Shuffle or change objects in the workspace
>
> 　　**end if**
>
> **end while**
>
> **return** $\mathcal{D}\_{play}$
>
> ---
> Phase 1 creates a safe pose library from 5 minutes of discovery teleoperation via voxel grid downsampling. Phase 2 autonomously generates trajectories by sampling waypoint sequences, enforcing contact-prone movements via a step density function, and connecting them through cosine interpolation with Gaussian noise. Human intervention is limited to periodic object shuffling (~every 30 minutes), highly scalable and substantially cheaper than expert demonstration collection.
>
> **Response to Weakness 3:**
>
> We sincerely apologize for the presentation issues. We will thoroughly proofread and reorganize the manuscript, ensuring consistent formatting, proper citations, and clearer structure.
>
> **Response to Weakness 4:**
>
> We conducted further experiments on complex sequential tasks: "Clean Table" and "Put & Push", fine-tuned with identical 200 trajectories:
>
> |Model|Clean Table (Part)|Clean Table (Ent)|Put & Push (Part)|Put & Push (Ent)|
> |-|-|-|-|-|
> |Baseline|25%|0%|35%|0%|
> |NORA|65%|30%|70%|40%|
> |TAP (Ours)|50%|30%|60%|40%|
>
> TAP exhibits competitive performance on multi-step tasks, demonstrating robustness compared to baselines.
>
> **Response to Minor Weakness: Action Space Inconsistency**
>
> Both simulation and real-world experiments use relative end-effector pose changes (delta EEF). We will clarify this in the revised manuscript.
>
> **Response to Key Question: Contribution of Existing vs. Autonomous Exploration Data**
>
> Our framework leverages two complementary sources: existing off-task trajectories and autonomous play . Both contribute to learning robust physical priors. Together, these results establish that task-agnostic data—whether from existing datasets or autonomous collection—provides significant benefits for learning transferable physical priors.
>
> Additional results and data collection details are available at https://anonymous.4open.science/r/TAP-icml.

---

> > ### Author Rebuttal · Reviewer_6cXk · 2026-04-03
> >
> > Thank you for the author's rebuttal. Adding more experiments in such a short period is not easy, which I really admire.
> >
> > I think the authors have addressed my previous concerns. For Q1, I now understand that the use of task-agnostic data enables zero additional annotation, as the collected data can be directly used after teleoperation or manipulation without requiring secondary labeling of task goals. This idea is somewhat related to Vision-Action (VA) models, which remove the reliance on language in traditional Vision-Language-Action (VLA) frameworks. In this context, the use of inverse kinematics (IK) is reasonable, and I am encouraged by the potential improvements in this direction.
> >
> > For Q2, I can see that task-agnostic data can contribute to the final performance. However, I also share the concern raised by other reviewers that the experiments are limited, making it difficult to fully assess the method's ability.
> >
> > In summary, the authors have addressed my concerns, and I believe the paper provides some meaningful new insights. I will consider raising my score in the final stage after reviewing the other reviewers’ comments.

---

> > > ### Author Response · Authors · 2026-04-06
> > >
> > > Dear Reviewer 6cXk,
> > >
> > > We sincerely appreciate your encouraging response and your recognition of the effort behind our additional experiments. We are glad that our rebuttal has addressed your major concerns, and we would like to take this opportunity to provide two further clarifications.
> > >
> > > ---
> > >
> > > **On the Full Scope of Task-Agnostic Data Collection**
> > >
> > > Your summary accurately captures the value of our simulation setting—repurposing existing teleoperation data requires zero additional annotation cost. We would like to further highlight an implication demonstrated in our real-world experiments that goes one step beyond this: the 30 hours of *autonomous random play* enables scalable data collection with almost **no human teleoperation at all**. Rather than simply reusing existing data, this paradigm allows robots to autonomously accumulate interaction experience with minimal human involvement. Our broader goal is to demonstrate that effective physical priors can be acquired without relying on costly human teleoperation, thereby reducing the annotation burden in VLA training.
> > >
> > > ---
> > >
> > > **On Experimental Scale and Engineering Constraints**
> > >
> > > We fully acknowledge the concerns regarding the current scale of our real-world evaluation, which we share with you and other reviewers. Scaling physical robot experiments introduces substantial engineering challenges that are difficult to convey in a paper. During our data collection and debugging phases, we encountered in total 4 servo failures (waist and shoulder joints) each of which take days to fix and recurring communication problems. We believe that as these engineering pipelines stabilize, our automated collection paradigm will enable rapid accumulation of large-scale data, yielding considerably more comprehensive results.
> > >
> > > To further address concerns regarding our method's capacity on complex tasks, we conducted an extended data-scaling ablation on "Clean the Table" task (20 trials each):
> > >
> > > | **Model** | **Partial** | **Entire** |
> > > |---|---|---|
> > > | **Stage 2: 200 Trajectories** | | |
> > > | Baseline | 25% | 0% |
> > > | NORA | 65% | 30% |
> > > | TAP (Ours) | 50% | 30% |
> > > | **Stage 2: 400 Trajectories** | | |
> > > | Baseline | 35% (+10%) | 5% (+5%) |
> > > | NORA | 70% (+5%) | 35% (+5%) |
> > > | TAP (Ours) | 65% (+15%) | **45% (+15%)** |
> > >
> > >
> > > As is shown in the table, scaling Stage 2 data from 200 to 400 trajectories yields substantially larger performance gains for TAP (+15% Partial, +15% Entire) relative to NORA (+5% Partial, +5% Entire), demonstrating that the learned physical representations are highly data-efficient and possess considerable capacity for further improvement.
> > >
> > > We demonstrate that when more fine-tuning data is involved, our proposed method shows a steeper performance gain compared to NORA, outperforming NORA in "Entire" Success rate.
> > >
> > > We will incorporate this analysis and the associated discussion of data distribution effects into the revised manuscript.
> > >
> > > ---
> > >
> > > We commit to carefully incorporating all clarifications, new ablations, and limitation discussions from this rebuttal into the revised manuscript. Thank you again for your constructive feedback and support.

---

### Official Review · Reviewer_Tdp7 · 2026-03-12

**Soundness:** 2
**Presentation:** 3
**Significance:** 3
**Originality:** 3
**Overall Recommendation:** 4
**Confidence:** 4

**Summary:**

This paper proposes a two-stage Task-Agnostic Pretraining (TAP) framework for Vision-Language-Action (VLA) models. The authors analyze a general topic of decoupled robotic learning. A broad domain considered by this paper is Embodied AI. The first stage uses unlabeled interaction data (e.g., discarded trajectories or random exploration) to learn physical affordances ("how to move") via an inverse dynamics objective (predicting the action from consecutive observations). The second stage aligns these physical priors with language instructions ("what to do") using minimal expert data via behavior cloning. Experiments are conducted on the SIMPLER benchmark and a real-world WidowX robot.

**Compliance With Llm Reviewing Policy:**

Affirmed.

**Final Justification:**

The author's response addressed my concerns. I will raise my score.

**Key Questions For Authors:**

Could you provide a direct comparison with other self-supervised representation learning methods (e.g., forward dynamics, video prediction, or Masked Autoencoders) under the exact same architecture? This would help clarify whether inverse dynamics is the optimal choice for this pretraining stage.

Given the current evaluation on two real-world tasks and 30 hours of exploration data, how well do you expect the learned physical priors to generalize to more complex, multi-step tasks or articulated object manipulation?

Regarding the comparisons in Tables 2 and 3, were the baseline models (e.g., OpenVLA, Octo) fine-tuned on the exact same subset of data as TAP? Clarifying this would help ensure a fair comparison and better highlight TAP's specific advantages.

**Limitations:**

It would be highly beneficial to include a dedicated Limitations section. For example, discussing potential failures of inverse dynamics under extremely high-frequency control, or addressing the inefficiency of pure random exploration for complex tasks, would provide a more balanced view of the work.

**Strengths And Weaknesses:**

Strengths:

Intuitive motivation: Decoupling policy learning into task-agnostic physical understanding and task-specific semantic alignment is a practical and logical approach to reduce the cost of expert data collection.

Improved robustness: Real-world experiments indicate that pretraining on random exploration data enhances the model's adaptability to visual distractors and distribution shifts.

Weaknesses:

Soundness: The scale of the experiments could be expanded to better support the core claims. Evaluating only two real-world tasks and four simulation tasks makes it challenging to fully validate the acquisition of a "universal physical prior." Additionally, the baselines could be more comprehensive.

Presentation: The writing is generally clear, but some experimental details—such as the exact diversity and coverage of the 30-hour real-world random exploration data—could be elaborated further to aid reproducibility.

Significance: Leveraging unlabeled data for pretraining is a valuable direction for VLA models. However, given the relatively small scale of the experiments, claims such as "breaking the data wall" might be slightly overstated.

---

> ### Author Rebuttal · Authors · 2026-03-31
>
> Dear Reviewer Tdp7,
>
> Thank you for your constructive and thorough review. Below, we systematically address your concerns:
>
>  **Response to W1: Soundness**
>
> We appreciate your critical insight regarding experimental scale. To address this, we have significantly expanded our evaluation across both simulation and real-world settings with diverse robot embodiments:
>
> **Simulation Google Robot Evaluation:**
>
> We evaluated TAP on Google Robot using the SIMPLER benchmark using fractal dataset with identical process:
>
> | Model | Move Near | Open/Close Drawer|  ||Open Top Drawer & Place Apple |
> |---|---|-|-|-|---|
> | | Average | Open | Close | Average | Average |
> | RT-1-X | 0.317 | 0.296 | 0.891 | 0.597 | 0.213 |
> | OpenVLA | 0.462 | 0.194 | 0.518 | 0.356 | 0 |
> | Baseline | 0.281 | 0.241 | 0.139 | 0.190 | 0 |
> | **TAP (Ours)** | **0.415** | **0.347** | **0.215** | **0.281** | **0.047** |
>
> **Real-World long-term Evaluation:**
>
>  We collected 200 additional trajectories for two complex sequential manipulation tasks:  "Cleaning the table" and "Put inside and Push Away", requiring coordinated grasping, placing, and manipulation over multiple steps. Models are evaluated over 10 trials:
>
> | Model | Clean Table (Partial) | Clean Table (Entire) | Put & Push (Partial) | Put & Push (Entire) |
> |---|---|---|---|---|
> | Baseline | 25% | 0% | 35% | 0% |
> | NORA | 65% | 30% | 70% | 40% |
> | **TAP (Ours)** | **50%** | **30%** | **60%** | **40%** |
>
> Our expanded evaluation demonstrates that TAP generalizes effectively across both different robot morphologies and complex real-world scenarios, despite using only 30 hours of task-agnostic pretraining data versus OXE's >1M expert trajectories.
>
> **Response to W1: Presentation**
>
> Thank you for highlighting the need for greater reproducibility details.  Across 30 hours of autonomous play, interactions distribute as follows: approximately **5%**grasping movements (grasp or object-in-gripper motions), 35% contact-rich interactions (pushing, pulling, or manipulation), and 60% free-space motions.  Our "random play" is procedurally constrained via a two-phase approach (detailed algorithm provided in reviewer FcMj's Q2 response and accompanying anonymous link ).
> We will expand this discussion in the revised manuscript and provide the accompanying materials for full reproducibility.
>
>  **Response to W1: Significance**
>
> We acknowledge that this work is positioned as an exploratory study addressing data constraints in VLA training. We will revise our language to avoid overstated claims such as "breaking the data wall" and iframe our contributions more precisely.
>
> **Response to Q1: Self-Supervised Objective Selection**
>
> To rigorously validate that Inverse Dynamics (ID) is the optimal choice for this pretraining stage, we conducted a direct comparison against Forward Dynamics (FD)—predicting the next visual embedding from current observation and action—under identical architecture and hyperparameters:
>
> |Model|Spoon on Cloth(Partial/Entire)|Carrot on Plate(Partial/Entire)|Stack Blocks(Partial/Entire)|Eggplant in Basket(Partial/Entire)|Avg-Partial|Avg-Entire|Avg-All|
> |---|---|---|---|---|---|---|---|
> |Baseline|41.67%/33.33%|48.00%/8.00%|37.50%/16.67%|0.00%/0.00%|31.79%|14.50%|23.15%|
> |ForwardDynamics|16.67%/12.50%|16.67%/0.00%|16.67%/0.00%|16.67%/0.00%|16.67%|3.13%|9.90%|
> |**TAP(ID)**|**66.7%/58.3%**|**50.0%/0.0%**|**58.3%/16.7%**|**8.3%/8.3%**|**45.82%**|**20.82%**|**33.32%**|
>
> Inverse Dynamics' superiority stems from structural alignment with the VLA paradigm rather than empirical convenience. A standard VLA maps (observation, language) → action; our inverse dynamics model maps (start observation, end observation) → action, treating the end observation as a "visual goal" analogous to language instructions. This symmetry enables seamless parameter sharing between the visual encoder and VLA backbone, ensuring that feature representations supporting language-conditioned prediction also support goal inference. Conversely, Forward Dynamics requires predicting high-dimensional future observations—a harder objective misaligned with the downstream task.
>
>  **Response to Q2: Generalization to Complex, Multi-Step Tasks**
>
> As demonstrated in our extended real-world evaluation, TAP generalizes effectively to multi-step sequential tasks. TAP achieves 50% success on partial outcomes and 30% on full task completion, matching NORA despite using 30 hours versus >1 million trajectories of pretraining data.
>
> **Response to Q3: Fair Baseline Comparisons**
>
> To clarify: RT-1-X and Octo results in simulation were directly sourced from the original SIMPLER benchmark paper. For all other evaluated baselines (OpenVLA, π₀, and NORA), we conducted fine-tuning using the identical Stage 2 expert data subset as TAP.
>
> Additional results, step judge function visualizations and data collection details are available at https://anonymous.4open.science/r/TAP-icml.
>
> We welcome any further questions and remain committed to addressing all concerns.

---

> > ### Author Rebuttal · Reviewer_Tdp7 · 2026-04-03
> >
> > Thank you for the substantial additions, including Google Robot experiments, multi-step real-world tasks, and the forward dynamics ablation, which collectively address most of my concerns. Two remaining points: (1) on Google Robot, TAP underperforms RT-1-X on drawer tasks (0.281 vs. 0.597) and falls behind OpenVLA on Move Near (0.415 vs. 0.462)—could you discuss when inverse dynamics pretraining may be less effective? (2) On the new multi-step tasks, TAP still lags behind NORA on partial success (50%/60% vs. 65%/70%)—does this suggest that task-agnostic physical priors are insufficient for complex sequential manipulation?

---

> > > ### Author Response · Authors · 2026-04-06
> > >
> > > Dear Reviewer Tdp7,
> > >
> > > We sincerely appreciate your continued engagement and your sharp follow-up questions. We are glad that the Google Robot evaluations, multi-step tasks, and Forward Dynamics ablation have addressed the majority of your concerns and we are happy to provide further clarification below.
> > >
> > > ---
> > >
> > > **Response to "TAP underperforms RT-1-X on drawer tasks (0.281 vs. 0.597) and falls behind OpenVLA on Move Near (0.415 vs. 0.462)" (Q1)**
> > >
> > > We apologize for not making the intended role of our baselines sufficiently clear.  Moreover, reference models including RT-1-X[1], OpenVLA, and NORA have all been pretrained on the Open X-Embodiment (OXE) dataset (1M+ language-annotated expert trajectories), whereas TAP uses only 5000 task-irrelevent trajectories in simulation or 30 hours of automated play in real world. These OXE models are therefore included as **approximate upper-bound references**, not as direct competitors. Our primary comparison target is the **no-pretraining baseline**.
> > >
> > > We will revise the evaluation section to clearly separate these baselines—the no-pretraining baseline (primary comparison), TAP (proposed method), and NORA (OXE upper-bound reference)—so that the performance gaps are interpreted in the appropriate context.
> > >
> > > ---
> > > **Response to "When inverse dynamics pretraining may be less effective?" (Q1)**
> > >
> > > We agree that the performance gaps on "Move Near" tasks deserve a more careful examination and it represents a typical failure mode where TAP being less effective.
> > >
> > > The Inverse Dynamics objective explicitly strips away language to focus on kinematics and physical causality, which means TAP relies entirely on the small finetuning dataset to build task-related language-action grounding. "Move Near" requires not only precise motor control but also correct identification and grounding of target objects from diverse instructions, making semantic understanding equally—if not more—critical than physical control, and placing it outside the sweet spot of ID pretraining. We will explicitly discuss this boundary in the revised Limitations section: **TAP is most effective for tasks bottlenecked by visual motor control and contact physics, and less so when the primary challenge is semantic reasoning.**
> > >
> > > ---
> > >
> > > **Response to "Does this suggest that task-agnostic physical priors are insufficient for complex sequential manipulation?" (Q2)**
> > >
> > >
> > > Thank you for this insightful question. We believe how we collect task-agnostic data has a strong connection to how the policy performs. For example, we currently use a safety-constrained random-play policy to collect data. This led to approximate 30 hours of random-play data consisting of  **~60% free-space motion, ~35% contact-rich interactions, and only ~5% precise grasping**. This distribution naturally leads to weaker initial grasp success compared to NORA, which is pretrained on dense expert grasping data.
> > >
> > > Rather than a flaw in the proposed task-agnostic pretraining paradigm, we view this limitation as an opportunity for improvement through the development of more advanced collection policies.
> > >
> > > Instead of simple random play, we could use policies that actively prioritize object manipulation. By intentionally collecting data that changes the environment (such as grasping and moving objects), we can provide the model with the dense interaction data it needs to master complex skills.
> > >
> > > ---
> > >
> > > Furthermore, we demonstrate that when more fine-tuning data is involved, our proposed method shows a steeper performance gain compared to NORA, outperforming NORA in "Entire" Success rate on "Clean the Table" task(20 trials each):
> > >
> > > | **Model** | **Partial** | **Entire** |
> > > |---|---|---|
> > > | **Stage 2: 200 Trajectories** | | |
> > > | Baseline | 25% | 0% |
> > > | NORA | 65% | 30% |
> > > | TAP (Ours) | 50% | 30% |
> > > | **Stage 2: 400 Trajectories** | | |
> > > | Baseline | 35% (+10%) | 5% (+5%) |
> > > | NORA | 70% (+5%) | 35% (+5%) |
> > > | TAP (Ours) | 65% (+15%) | **45% (+15%)** |
> > >
> > >
> > > As is shown in the table, scaling Stage 2 data from 200 to 400 trajectories yields substantially larger performance gains for TAP (+15% Partial, +15% Entire) relative to NORA (+5% Partial, +5% Entire), demonstrating that the learned physical representations are highly data-efficient and possess considerable capacity for further improvement.
> > >
> > > We will incorporate this analysis and the associated discussion of data distribution effects into the revised manuscript.
> > >
> > > ---
> > >
> > > Thank you again for highlighting these aspects; it has been very helpful for improving the paper. We remain open to any further discussion.
> > >
> > > [1] Evaluating Real-World Robot Manipulation Policies in Simulation, Conference on Robot Learning, 2024

---

### Official Review · Reviewer_FcMj · 2026-03-13

**Soundness:** 3
**Presentation:** 3
**Significance:** 2
**Originality:** 2
**Overall Recommendation:** 3
**Confidence:** 5

**Summary:**

The paper proposes Task-Agnostic Pretraining (TAP), a simple two-stage recipe for training Vision-Language-Action (VLA) policies: first, pretrain on unlabeled, task-agnostic robot interaction data using an inverse dynamics objective to learn “how to move,” then fine-tune with a small set of language-annotated expert demonstrations to learn “what to do.” Empirically, the authors report gains over training from scratch on the SIMPLER benchmark and on two real-robot tasks, with particular improvements in low-level affordance metrics and in robustness to real-world visual perturbations.

**Compliance With Llm Reviewing Policy:**

Affirmed.

**Final Justification:**

I will maintain my score as my concerns are not fully resolved.

**Key Questions For Authors:**

1. Did you explore multi-step conditioning or action horizons, and how sensitive are the results to k and to the downsampling rate?
2. Can you provide more details about the autonomous play data: exploration policy, object diversity, safety constraints, and any failure filtering? How much of the 30h play involved contact-rich interactions vs free-space motion?

**Limitations:**

The authors should discuss whether the uniqueness of actions in the inverse dynamics stage limits the multimodal expressiveness of VLA-generated actions.

**Strengths And Weaknesses:**

Strengths:
1. The paper advances a clear and compelling decomposition hypothesis: decouple motor priors (“how to move”) from semantic alignment (“what to do”), and learn the former from cheap task-agnostic data.
2. Using inverse dynamics as a standalone pretraining phase for VLA backbones is simple, broadly applicable, and easy to reproduce, making the contribution accessible and potentially impactful for practitioners with limited labeled data.
3. The methodology is presented concisely; the training objectives and action space are easy to follow.

Weaknesses:
1. The inverse dynamics pretraining is a known idea, making the methodological novelty modest; the main contribution is an application and empirical study rather than a fundamentally new algorithm.
2. The paper states that inverse dynamics “forces the visual encoder to learn dynamics-aware representations,” yet Implementation Details say the visual encoder is frozen in Stage 1; this is an internal inconsistency that undermines the claimed mechanism.
3. Real-world evaluation is limited to two tasks with relatively small sample sizes and mixed overall averages (e.g., TAP underperforms NORA on the carrot task average), making it hard to generalize broad claims.

---

> ### Author Rebuttal · Authors · 2026-03-31
>
> Dear Reviewer FcMj,
>
> Thank you for your thorough review and insightful feedback. We address your concerns systematically below.
>
> **Response to W1: Methodological Novelty**
>
> While Inverse Dynamics Models are well-established, our contribution lies in demonstrating how cheap, task-agnostic data—typically underutilized and discarded—serves as an effective pretraining foundation for VLAs. IDM acts as a self-supervised mechanism to extract physical priors from unstructured interactions. The core innovation is the data scaling paradigm and the principled decoupling of physical grounding from semantic alignment. We will clarify this positioning in the revised manuscript.
>
> **Response to W2: Visual Encoder Training**
>
> We apologize for the inconsistency. The vision encoder is **trained** in both stages, not frozen. This will be corrected in the revision to accurately reflect our methodology.
>
> **Response to W3: Real-World Evaluation & NORA Comparison**
>
> We collected 200 additional trajectories for two complex sequential tasks: "Cleaning the table" and "Put inside and Push Away" . All models are fintuned using same amount of data and evaluated over 10 trials:
>
> |Model|Clean Table (Part)|Clean Table (Ent)|Put & Push (Part)|Put & Push (Ent)|
> |-|-|-|-|-|
> |Baseline|25%|0%|35%|0%|
> |NORA|65%|30%|70%|40%|
> |TAP (Ours)|50%|30%|60%|40%|
>
> Regarding the carrot task performance: NORA was included to compare TAP pretraining (30h) against the Open X-Embodiment dataset (>1M trajectories). As shown in the table, TAP remains competitive in complex tasks and outperforms NORA under severe visual distribution shifts, demonstrating the robustness of our approach.
>
> **Response to Q1: Multi-Step Conditioning & Temporal Offset**
>
> We did not employ a sliding multi-step horizon and applyed a fixed offset $k$. As detailed in Section 3.4, raw trajectories at 25Hz exhibit minimal visual displacement between frames, making inverse dynamics ill-posed. We fixed downsampling to 5Hz to ensure meaningful visual changes, thus conditioning on observation pairs separated by a fixed 0.2-second interval.
>
> **Response to Q2: Autonomous Play Details, Safety Constraints & Contact Ratio**
>
> Our "random play" is procedurally constrained, not uniformly random. We employ a two-phase approach:
>
> ---
> **Algorithm: Constrained Procedural Trajectory Generation**
>
> **Require:** Raw teleoperation joint poses $\mathcal{P}\_{raw}$, safety bounds $\mathcal{B}$, voxel size $v\_{size}$, maximum waypoint distance $d$, density function $\mathcal{p}\_{interact}$ ensures interaction, noise scale $\sigma$, Forward Kinematic function $FK$.
>
> **Ensure:** Procedural trajectory dataset $\mathcal{D}\_{play}$
>
> **Phase 1: Safe Pose Library Initialization**
>
> $\mathcal{P}\_{safe}\leftarrow\\{ p \mid p \in\mathcal{P}\_{raw}\cap\mathcal{B}, P \in\mathbb{R}^{dof} \\}$
>
> $\mathcal{P}\leftarrow\text{VoxelDS}(\mathcal{P}\_{safe}, v\_{size}, FK)$ *(Note: VoxelGridDownsample in EEF)*
>
> **Phase 2: Autonomous Data Collection**
>
> $\mathcal{D}\_{play}\leftarrow\emptyset$
>
> **while** True **do**
>
> 　　$\mathcal{W}\sim\\{ (p\_{1}, \dots, p\_{\tau}) \mid p\_{i} \in\mathcal{P} \, \|FK(p\_{i}) - FK(p\_{i+1})\|\leq d\, \mathcal{E}\\lt \mathcal{p}_{interact}(FK\_{z}(p\_{i}))\ where\ \mathcal{E} \sim\mathcal{U}(0, M) \\}$ *(Reject Sampling)*
>
> 　　$\mathcal{W}\leftarrow CosineInterpolate(\mathcal{W}) \in\mathbb{R}^{n \times dof}$
>
> 　　$\mathcal{W}\sim\\{ (w\_{1}, \dots, w\_{n})\mid w\_{i} \sim\mathcal{N}(p\_{i}, \sigma) \, \forall p\_{i} \in\mathcal{W} \, w\_{i}\in\mathcal{B} \\}$
>
> 　　$\mathcal{D}\_{play}\leftarrow\mathcal{D}\_{play} \cup \\{ (\tau, o) \mid o \leftarrow\text{exec}(\tau), \tau \in\mathcal{W} \\}$
>
> 　　**if** human intervention triggered (e.g., $\Delta t \geq 30$ mins) **then**
>
> 　　　　Shuffle or change objects in the workspace
>
> 　　**end if**
>
> **end while**
>
> **return** $\mathcal{D}\_{play}$
>
> ---
> **Phase 1** creates a safe pose library from 5 minutes of operator teleoperation, filtered via voxel grid downsampling to ensure uniform coverage. **Phase 2** autonomously generates trajectories in three environments by sampling waypoint sequences from the safe library, enforcing contact-prone movements via a step density function with higher acceptance to lower end-effector positions, and connecting them through cosine interpolation with Gaussian noise. Human intervention is limited to periodic object shuffling (~every 30 minutes).
>
> Across collected autonomous play data, the interaction breakdown is as follows: approximately **5%** grasping movements (grasp or object-in-gripper motions), **35%** contact-rich interactions (pushing, pulling, or manipulation), and **60%** free-space motions. This distribution reflects realistic robot behavior and provides a rich foundation for learning robust physical priors.
>
> Additional results, step judge function visualizations and data collection details are available at https://anonymous.4open.science/r/TAP-icml.
>
> We hope our responses solve your concerns and we welcome any further discussion.

---

> > ### Author Rebuttal · Reviewer_FcMj · 2026-04-03
> >
> > Although the author's rebuttal addressed some of the weaknesses and questions I raised, I still maintain my own view regarding the novelty of the method (W1). Moreover, in the author's response to W3, the experimental results still show a gap with NORA on certain metrics.

---

> > > ### Author Response · Authors · 2026-04-06
> > >
> > > Dear Reviewer FcMj,
> > >
> > > We sincerely appreciate your continued engagement. We are glad that concerns regarding the visual encoder inconsistency (W2), multi-step conditioning (Q1), autonomous play details (Q2), and limitations have been largely resolved. The two remaining points on methodological novelty (W1) and the NORA performance gap (W3) are well-taken, and we would like to address them directly below.
> > >
> > > ---
> > >
> > > **On Methodological Novelty (W1)**
> > >
> > > We acknowledge that our current presentation may have foregrounded the IDM algorithm and we sincerely apologize for the confusion.
> > >
> > > We would like to be direct: our primary contribution is not the invention of the Inverse Dynamics pretraining, but **the proposal and validation of Task-Agnostic Pretraining (TAP)** as a new data-scaling paradigm.
> > >
> > > The contribution of TAP is that it aims to address a fundamental bottleneck in current VLA training—the overreliance on expensive human teleoperation. It validates the *Decomposition Hypothesis* by decoupling learning into two steps: first using massive amounts of cheap, unlabeled data (like random play) to learn physical skills ("how to move"), and then using a tiny amount of expensive expert data to learn the task ("what to do").
> > >
> > > Inverse Dynamics serves purely as our chosen mechanism to implement this pretraining effectively, not as the contribution itself.
> > >
> > > We will revise the introduction and abstract to explicitly highlight Task-Agnostic Pretraining (TAP) as our core contribution, clearly stating that Inverse Dynamics is just one effective way to implement it.
> > >
> > > ---
> > >
> > > **On the Performance Gap with NORA (W3)**
> > >
> > > First, we sincerely apologize for the confusion caused by our unclear explanation of the experimental setup and model comparaisons.
> > >
> > > We will clarify in the revised version about the intended role of NORA in our evaluation. Models including NORA and OpenVLA have been pretrained on the **Open X-Embodiment (OXE) dataset**, comprising over **1 million** expert-teleoperated, language-annotated trajectories—representing an enormous investment of human labor and annotation cost. TAP, by contrast, uses only **30 hours of autonomous random play** requiring minimal human effort as its pretraining data.
> > >
> > > Therefore, our primary experimental comparison is therefore **TAP vs. the no-pretraining baseline**, which proves the value of task-agnostic pretraining. We included NORA not as a direct competitor, but as an **approximate upper-bound reference** to show what large-scale, high-quality expert pretraining can achieve. Against this high reference, the fact that TAP achieves *comparable overall performance* and *superior robustness in unseen environments* (with much less annotation effort) proves the huge potential of our method.
> > >
> > > We will revise the evaluation section to clearly separate these baselines—the no-pretraining baseline (primary comparison), TAP (proposed method), and NORA (OXE upper-bound reference)—so that the performance gaps are interpreted in the appropriate context.
> > >
> > > ---
> > >
> > > We hope these clarifications better convey the scope of our contribution and remain open to further discussion.

---

### Official Review · Reviewer_CKXk · 2026-03-13

**Soundness:** 3
**Presentation:** 4
**Significance:** 3
**Originality:** 3
**Overall Recommendation:** 5
**Confidence:** 4

**Summary:**

The paper introduces Task-Agnostic Pretraining (TAP), a two-stage VLA pretraining framework that first pretrains on abundant, cheap task-agnostic data in a self-supervised manner to learn physical affordances, then pretrains on minimal expert data to align physical priors with language instructions. The experiments in both simulation and the real world show that TAP learns robust, transferable physical representations for Embodied AI.

**Compliance With Llm Reviewing Policy:**

Affirmed.

**Final Justification:**

The rebuttal provides additional experiments on multi-embodiment generalization and a detailed explanation of the constrained procedural trajectory generation, which effectively addressed my concerns. The scaling results for autonomous data further strengthen the paper's claims. I'll raise my score accordingly.

**Key Questions For Authors:**

1. Could you provide details about collecting the 30-hour autonomous random play? If the robot executes truly random actions, it is highly likely to wave aimlessly in free space or trigger safety stops by colliding with the table. How did you constrain or sample these actions to ensure "meaningful" interactions with objects while maintaining operational safety?
2. How exactly were the task-specific expert data trajectories collected and sampled? Were these 200 real-world data gathered via standard teleoperation? What specific distribution of behaviors do the 5k simulation expert data cover?
3. In Figure 4, 70k Stage 1 pretraining steps unexpectedly outperform 80k and 90k steps across various Stage 2 intervals. What is the underlying cause of this performance degradation at higher pretraining steps? Additionally, although full results are in the appendix, including the heatmaps of avg-partial and avg-entire would better support the claims regarding the distinct functions of stage 1 and stage 2.

**Limitations:**

Yes.

**Strengths And Weaknesses:**

Pros:

1. The paper is very well-written and logically structured. The core hypothesis is well-motivated, the method is sound, and the conclusions drawn from the results are insightful.
2. The training paradigm is elegant, smartly reusing the VLM backbone and action head across both the task-agnostic and task-specific training stages.
3. The SIMPLER benchmark experiments demonstrate that the TAP framework yields performance gains while requiring less expert data than competing VLA models. The real-world experiments further showcase the model's robustness against varying backgrounds and novel camera viewpoints.

Cons:

1. The paper emphasizes that the task-agnostic stage learns affordances from self-exploration trajectories, avoiding the bottleneck of human demonstrations. However, it fails to explain the concrete methodology behind this process. Specifically, the details on how to safely collect high-quality, meaningful "random play" data are missing. If actions are truly random, the robot would likely spend most of its time executing meaningless movements in the air without interacting with objects, or it would trigger safety stops by colliding with the table.
2. While the paper provides data scaling ablations for the simulation benchmark, it does not ablate the volume or diversity of the real-world self-exploration data (the 30 hours of random play), leaving it unclear how much autonomous data is actually required to achieve the claimed robustness.
3. The evaluation is restricted to only 4 simulation tasks and 2 real-world tasks. Additionally, the embodiment (WidowX) remains consistent across both the training data (Bridge dataset, real random-play) and the evaluation settings (SIMPLER and real-world exp). This narrow scope fails to adequately support the paper's broader claims regarding multi-embodiment generalization and scalability for general-purpose Embodied AI.
4. Citation error (?) in Line 665.

---

> ### Author Rebuttal · Authors · 2026-03-31
>
> Dear Reviewer CKXk
>
> We sincerely thank you for your thoughtful review and for recognizing the elegance of our training paradigm . Below we address your concerns point by point.
>
> Response to Q1 & W1:
>
> Our "Random Play" is procedurally constrained for safety and contact-rich interactions:
>
> ---
>
> **Algorithm: Constrained Procedural Trajectory Generation**
>
> **Require:** Raw teleoperation joint poses $\mathcal{P}\_{raw}$, safety bounds $\mathcal{B}$, voxel size $v\_{size}$, maximum waypoint distance $d$, density function $\mathcal{p}\_{interact}$ ensures interaction, noise scale $\sigma$, Forward Kinematic function $FK$.
>
> **Ensure:** Procedural trajectory dataset $\mathcal{D}\_{play}$
>
> **Phase 1: Safe Pose Library Initialization**
>
> $\mathcal{P}\_{safe} \leftarrow \\{ p \mid p \in \mathcal{P}\_{raw} \cap \mathcal{B}, P \in \mathbb{R}^{dof} \\}$
>
> $\mathcal{P} \leftarrow \text{VoxelDS}(\mathcal{P}\_{safe}, v\_{size}, FK)$ *(Note: VoxelGridDownsample in EEF)*
>
> **Phase 2: Autonomous Data Collection**
>
> $\mathcal{D}\_{play}\leftarrow\emptyset$
>
> **while** True **do**
>
> 　　$\mathcal{W}\sim\\{ (p\_{1}, \dots, p\_{\tau}) \mid p\_{i} \in \mathcal{P} \, \|FK(p\_{i}) - FK(p\_{i+1})\| \leq d\,\mathcal{E} \\lt \mathcal{p}_{interact}(FK\_{z}(p\_{i}))\ where\ \mathcal{E} \sim\mathcal{U}(0, M) \\}$ *(Reject Sampling)*
>
> 　　$\mathcal{W}\leftarrow CosineInterpolate(\mathcal{W}) \in \mathbb{R}^{n \times dof}$
>
> 　　$\mathcal{W}\sim\\{ (w\_{1}, \dots, w\_{n}) \mid w\_{i} \sim\mathcal{N}(p\_{i}, \sigma) \, \forall p\_{i} \in\mathcal{W} \, w\_{i} \in \mathcal{B}\\}$
>
> 　　$\mathcal{D}\_{play} \leftarrow\mathcal{D}\_{play}\cup\\{ (\tau, o) \mid o \leftarrow\text{exec}(\tau), \tau \in \mathcal{W} \\}$
>
> 　　**if** human intervention triggered (e.g., $\Delta t \geq 30$ mins) **then**
>
> 　　　　Shuffle or change objects in the workspace
>
> 　　**end if**
>
> **end while**
>
> **return** $\mathcal{D}\_{play}$
>
> ---
>
> **Phase 1** creates a safe pose library from 5 minutes of operator teleoperation, filtered via voxel grid downsampling to ensure uniform coverage. **Phase 2** autonomously generates trajectories by sampling waypoint sequences from the safe library, enforcing contact-prone movements via a step density function that assigns higher acceptance probabilities to lower end-effector positions, and connecting them through cosine interpolation with Gaussian noise. Human intervention is limited to periodic object shuffling (~every 30 minutes). This procedurally ensures contact-rich interactions while maintaining safety.
>
> Response to W2: Real-World Data Scaling Ablation
> As shown below, downstream performance strictly scales with autonomous data volume, confirming our hypothesis. (Heatmaps are provided in the anonymous link).
> |Stage1steps(k)|20(Stage2)|40(Stage2)|60(Stage2)|80(Stage2)|100(Stage2)|
> |-|-|-|-|-|-|
> |100|20%|45%|55%|70%|75%|
> |80|20%|35%|60%|55%|65%|
> |60|5%|20%|35%|30%|45%|
> |40|0%|10%|5%|25%|30%|
> |20|0%|0%|5%|20%|25%|
>
> # Response to W3: Multi-Embodiment Generalization and Complex Tasks
>
> We agree that evaluating solely on WidowX limits multi-embodiment generalization claims. To address this, we conducted additional experiments:
>
> **1. Cross-Embodiment Evaluation (Google Robot on SIMPLER):**
> We evaluated TAP on Google Robot using the Fractal2022 dataset with identical task-agnostic training logic.
>
> |Model|Move Near|Open/Close Drawer (Open) |Open/Close Drawer (Close) | Open/Close Drawer (Avg) | Open Top Drawer & Place Apple |
> |-|-|-|-|-|-|
> |RT-1-X|0.317|0.296|0.891|0.597|0.213|
> |OpenVLA|0.462|0.194|0.518|0.356 |0.000|
> |Baseline|0.281|0.241|0.139|0.190|0.000|
> |TAP (Ours)|0.415|0.347|0.215|0.281|0.047|
>
> **2. Complex Long-Horizon Tasks (Real-World):**
> We conducted further experiments on complex sequential tasks: "Clean Table" and "Put & Push":
>
> |Model|Clean Table(Part)|Clean Table(Ent)|Put&Push(Part)|Put&Push(Ent)|
> |-|-|-|-|-|
> |Baseline|25%|0%|35%|0%|
> |NORA|65%|30%|70%|40%|
> |TAP(Ours)|50%|30%|60%|40%|
>
> TAP transfers robustly across kinematics and visual domains, and handles multi-step sequential tasks effectively.
>
> **Response to Q2: Data Collection Details**
>
> *Simulation (Bridge):* Stage 1 uses 20k randomly sampled task-agnostic trajectories strictly disjoint from evaluation tasks. Stage 2 uses 5k randomly sampled task-specific trajectories, maintaining the original Bridge dataset's environmental distribution.
>
> *Real-World:* Stage 1 consists of 30 hours of autonomous play initialized by a 5-minute human pose library. Stage 2 uses 200 expert trajectories collected via standard human teleoperation on target tasks.
>
> **Response to Q3: Performance Anomaly at 70k Steps**
>
> In low expert-data regimes, policies are susceptible to training variance and occasional convergence to suboptimal local minima. However, the macro-trend clearly demonstrates that scaling Stage 1 data raises the performance ceiling.
>
> Additional results, step density function, heatmap visualizations for question 4, and data collection details are available at https://anonymous.4open.science/r/TAP-icml.

---

> > ### Author Rebuttal · Reviewer_CKXk · 2026-04-02
> >
> > Thanks for the thorough rebuttal. The additional experiments on multi-embodiment generalization and the detailed explanation of the constrained procedural trajectory generation effectively address my concerns. The scaling results for autonomous data further strengthen the paper's claims. I will raise my score accordingly.

---

> > > ### Author Response · Authors · 2026-04-03
> > >
> > > Dear Reviewer CKXk,
> > >
> > > We would like to sincerely thank you for your time and the positive acknowledgment of our rebuttal. We are encouraged to see that the additional experiments on multi-embodiment generalization and the detailed explanation of the procedural trajectory generation successfully addressed your concerns.
> > >
> > > We also truly appreciate your decision to raise the score. Your insightful comments have significantly helped us improve the clarity and quality of our work. We will ensure that the added details and experimental results are properly integrated into the revised version of the manuscript.
> > >
> > > Best regards,
> > >
> > > The Authors

---

### Decision · Program_Chairs · 2026-04-30

**Decision:**

Accept (regular)

**Comment:**

The paper proposes Task-Agnostic Pretraining (TAP), which first trains a VLA backbone on task-agnostic data via an inverse dynamics objective, and then finetunes it using a small amount of task-annotated data. The proposed idea is a strong and timely contribution. The core idea is simple, well motivated, and practically important for reducing dependence on expensive expert-labeled VLA data. The results are especially compelling in showing that inexpensive task-agnostic interaction data can provide useful and transferable physical representations.

Reviewers agreed that the paper presents a clear and elegant training paradigm, and the rebuttal substantially strengthened the empirical case by clarifying the autonomous data collection process, adding scaling analyses, cross-embodiment experiments, and more challenging real-world tasks. In particular, the results on SIMPLER and the real-world robustness experiments suggest that TAP provides a meaningful improvement over standard behavioral cloning while using far less labeled data, which I view as the main contribution of the paper.

The main limitations concern the modest methodological novelty of inverse-dynamics pretraining itself and the still limited scale of the experiments relative to the broadest claims about scalability or universal physical priors. However, I do not think the paper's value depends on introducing a fundamentally new algorithm. Rather, its contribution is in validating a useful and scalable training recipe, and the rebuttal makes this framing much clearer. The additional experiments also address many of the soundness concerns raised in the weaker reviews.

Overall, I believe this paper provides an insightful and practically relevant contribution to robot learning and VLA training, with enough empirical support to merit acceptance.